# Angiopoietin-4-dependent venous maturation and fluid drainage in the peripheral retina

Harri Elamaa[1,2†], Minna Kihlström[1,2†], Emmi Kapiainen[1,2], Mika Kaakinen[1,2], Ilkka Miinalainen[2], Symantas Ragauskas[3], Marc Cerrada-Gimenez[3], Satu Mering[3], Marjut Nätynki[1,2], Lauri Eklund[1,2]*

[1]Oulu Center for Cell-Matrix Research, Faculty of Biochemistry and Molecular Medicine, University of Oulu, Oulu, Finland; [2]Biocenter Oulu, University of Oulu, Oulu, Finland; [3]R&D Department, Experimentica Ltd, Kuopio, Finland

**Abstract** The maintenance of fluid homeostasis is necessary for function of the neural retina; however, little is known about the significance of potential fluid management mechanisms. Here, we investigated angiopoietin-4 (Angpt4, also known as Ang3), a poorly characterized ligand for endothelial receptor tyrosine kinase Tie2, in mouse retina model. By using genetic reporter, fate mapping, and in situ hybridization, we found *Angpt4* expression in a specific sub-population of astrocytes at the site where venous morphogenesis occurs and that lower oxygen tension, which distinguishes peripheral and venous locations, enhances Angpt4 expression. Correlating with its spatiotemporal expression, deletion of *Angpt4* resulted in defective venous development causing impaired venous drainage and defects in neuronal cells. In vitro characterization of angiopoietin-4 proteins revealed both ligand-specific and redundant functions among the angiopoietins. Our study identifies Angpt4 as the first growth factor for venous-specific development and its importance in venous remodeling, retinal fluid clearance and neuronal function.
DOI: https://doi.org/10.7554/eLife.37776.001

*For correspondence:
lauri.eklund@oulu.fi

†These authors contributed equally to this work

Competing interests: The authors declare that no competing interests exist.

## Introduction

Maintenance of fluid homeostasis is necessary for normal functions of the retina. Defects in supporting mechanisms result in macular edema that is the major cause of vision loss in common vascular and inflammatory diseases including diabetes, retinal vein occlusions and neovascularization affecting millions of people globally (*Daruich et al., 2018*). While vascular barrier disruption resulting in excessive leakage has been extensively studied, much less is known about the importance of fluid-removal mechanisms of neural retina that lacks lymphatic draining system.

The endothelial receptor tyrosine kinases Tie1 and Tie2 and the angiopoietin ligands (Angpt1–4) form a signaling pathway for blood and lymphatic vessel development (*Eklund et al., 2017*). Angiopoietin family studies thus far have focused on Angpt1 and Angpt2. As a significant limitation, the physiological importance of mouse Angpt4 (previously also named as Ang3 [*Kim et al., 1999*], Angpt4 in accordance with the HUGO Gene Nomenclature Committee and Mouse Genome Informatics) and its human orthologue ANGPT4 (*Valenzuela et al., 1999*) have remained unknown. No gene-targeted model for Angpt4 has been published and ANGPT4 has not yet been linked to any human disease.

Thus far, controversial results regarding Angpt4 functions have been reported showing action as a Tie2 agonist or antagonist in vitro (*Kesler et al., 2015*; *Lee et al., 2004*; *Valenzuela et al., 1999*). In mouse, excess of exogenous Angpt4 or ANGPT4 similarly induced blood and lymphatic vascular remodeling (*Kim et al., 2007*; *Lee et al., 2004*) and ANGPT4 improved vascular function in

streptozotocin-induced model for diabetes (*Kwon et al., 2013*), suggesting that Angpt4, ANGPT4 and Angpt1 have similar functions. In mouse tumor transplantation studies, exogenous Angpt4 has been reported to either inhibit (*Xu et al., 2004*) or promote tumor growth (*Brunckhorst et al., 2010*; *Kesler et al., 2015*), and similarly to Angpt2, Angpt4 expression has been reported to increase in hypoxia (*Abdulmalek et al., 2001*).

To clarify the physiological importance of Angpt4, we generated independently targeted mouse alleles to investigate cellular source, regulation of expression and the effect of Angpt4 deficiency in mouse retina that provides a widely used model for blood vessel growth, remodeling, maturation and vascular pathologies. In addition, *Angpt1* (*Lee et al., 2013*), *Angpt2* (*Gale et al., 2002*), *Tie1* (*D'Amico et al., 2014*), and *Tie2* (*Chu et al., 2016*) deletions are thoroughly investigated in postnatal mouse retina providing a comprehensive reference for assessing Angpt4 in vivo functions among the angiopoietins. Pathophysiological relevance of Angpt4 deficiency was evaluated in oxygen-induced retinopathy (OIR) model and using histopathological and ultrastructural analysis of postnatal and aged mice. Visual and venous functions were investigated using flash electroretinography and fluorescent tracers. We found Angpt4 expression in a specific population of hypoxia-regulated astrocytes that were enriched in the peripheral segment of the retina and locating close to the developing veins. Correlating with the strictly regulated expression pattern, genetic deletion of Angpt4 resulted in defective venous development and alterations in neural retina in adult mice secondary to impaired venous remodeling. Angpt4 deficiency did not affect capillaries or arteries either in physiological development, during aging or in retinopathy in OIR model, indicating a venous-specific function. Comparison of biochemical properties and cellular responses of Angpt4 and ANGPT4 to those of ANGPT1 and ANGPT2 provided novel mechanistic insights into the roles of Angpt4 and ANGPT4 and indicated both ligand-specific and redundant functions among the angiopoietins. Collectively, we identify Angpt4 as the first growth factor having a vessel-type-specific effect on venous development. Our data also reveals functional importance of a specific vein type in the peripheral retina, novel aspects of the complex Angpt/Tie pathway and complementary roles for angiopoietins in the establishment of the retinal circulatory system.

## Results

### Angpt4 is expressed in a distinct population of glial cells located close to the developing veins in the peripheral segment of postnatal mouse retina

In mice, the primary capillary plexus reaches the retinal periphery approximately at postnatal day (P) 8. Vascular remodeling and arteriovenous differentiation occur radially from the optic nerve head and different vessel types can be distinguished based on their morphology at P3 (*Crist et al., 2017*; *Stahl et al., 2010*). To investigate Angpt4 expression and its physiological importance, we generated targeted *Angpt4*LacZ and *Angpt4*Cre mouse alleles and crossed *Angpt4*Cre with a genetic fate mapping line *Rosa26*mT/mG (*Figure 1—figure supplement 1A–C*). In whole mount retinal preparations, Angpt4 expressing cells were strongly enriched by P9 in *Angpt4*LacZ mice (*Figure 1A*), at the time when peripheral venous maturation occurs (*Crist et al., 2017*). At P20 (*Figure 1B*), when maturation of retinal vasculature is nearly completed, Angpt4 expressing cells were decreased in number and thereafter only few were observed close to the veins and perivenous capillaries. The same expression pattern was detected in *Angpt4*Cre; *Rosa26*mT/mG mouse retina (*Figure 1C–G*), but not in controls (*Figure 1—figure supplement 2A–E*). During venous development, in both independent mouse lines, Angpt4 expressing cells were not randomly positioned but preferentially located in the peripheral segments while central retina was virtually negative (*Figure 1E* and *Figure 1—figure supplement 2C*).

Based on their morphology, position in the superficial retinal layer, and specific cell marker gene (GFAP) expression, we identified Angpt4 expressing cells as astrocytes (*Figure 2A–D*). During the postnatal development, vasculature is expanded and perfused from the optic nerve head towards the peripheral retina creating an oxygen gradient resembling Angpt4 expression pattern. In addition, Angpt4+ astrocytes were less numerous close to arteries (*Figure 2A–B*) where oxygen tension is higher than in veins. To confirm that low oxygen level positively regulates Angpt4 expression, *Angpt4*Cre/+; *Rosa26*mT/mG mice were exposed to low oxygen atmosphere (*Figure 2E–G*) and retinal

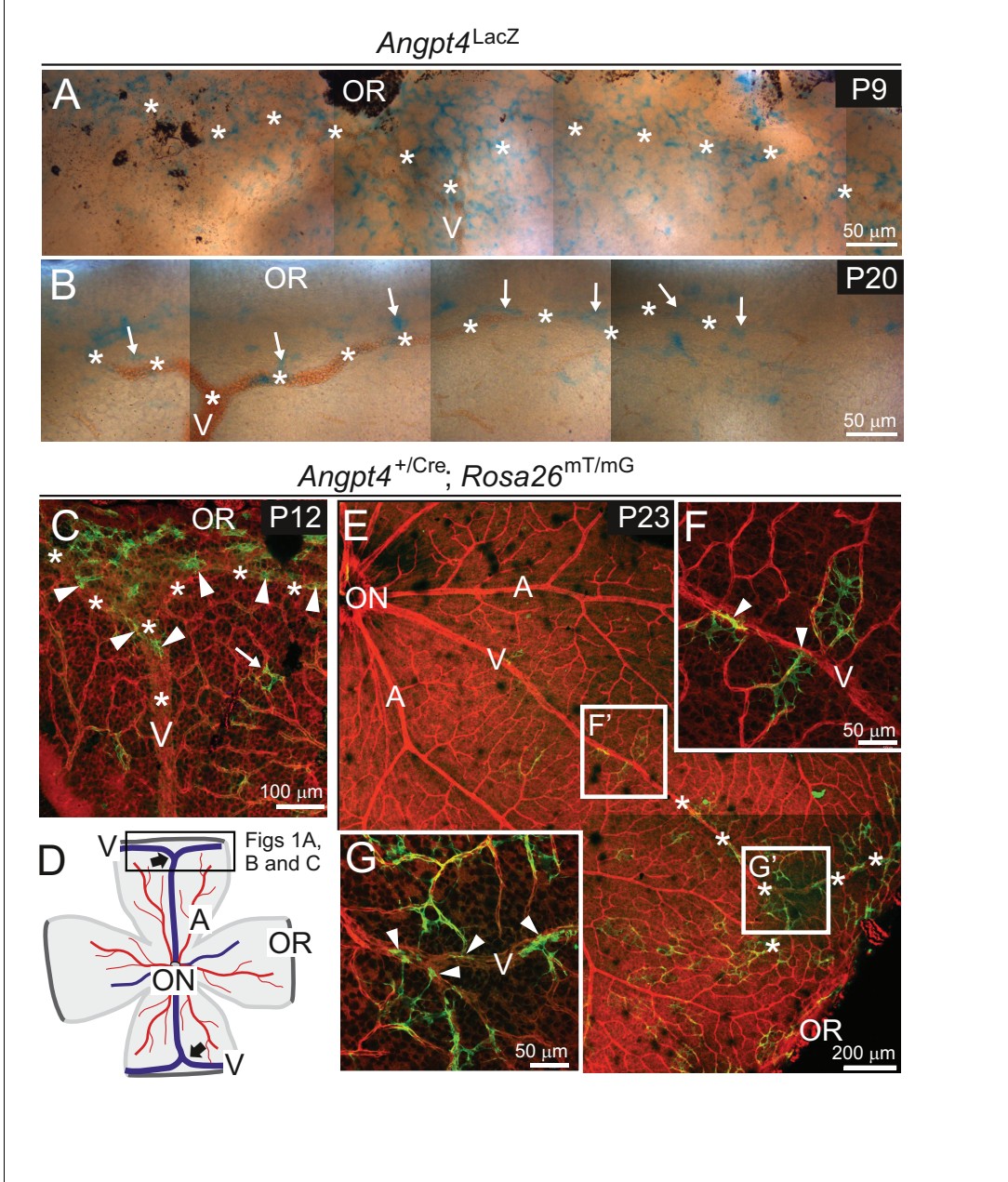

**Figure 1.** Angpt4 expression is temporally and spatially regulated at venous development sites in the peripheral retina. (A–G) Whole-mount retinas from *Angpt4*<sup>LacZ</sup> and *Angpt4*<sup>+/Cre</sup>; *Rosa26*<sup>mT/mG</sup> mice at the indicated postnatal days (P). (A) *Angpt4*<sup>LacZ</sup> allele visualized with X-Gal staining (blue) indicated numerous Angpt4 expressing cells in the P9 peripheral retina at the sites of developing vein (V, asterisks) as identified based on anatomical location, morphology and reticulocytes. (B) *Angpt4*<sup>LacZ</sup> allele revealed continued but lower Angpt4 expression (blue, arrows) around matured vein (V) in peripheral retina at P20. (C) In *Angpt4*<sup>+/Cre</sup>; *Rosa26*<sup>mT/mG</sup> mice, mTomato (red) is ubiquitously expressed until *Angpt4*-promoter-driven Cre-mediated loxP recombination induces GFP (green) expression. P12 retina shows GFP-positive cells preferentially locating in the peripheral part of retina close to the branching vein (arrowheads). Arrow indicates rare single GFP-positive cells that are apart from vein (asterisks). Angpt4-expressing cells in *Angpt4*<sup>+/Cre</sup>; *Rosa26*<sup>mT/mG</sup> mice are labeled permanently, that is they remain GFP positive even if Angpt4 expression is discontinued and may represent a progeny of Angpt4-expressing cells. (D) A schematic representation of mouse retina preparation. Peripheral venous circulation develops typically as two Y-shaped veins (thick arrows) starting after the first postnatal week and branching in the peripheral segment to form annular structure. Vein (V, blue line); artery (A, red line); OR, ora serrata; ON, optic nerve head; frame, location of microscopic analysis in *Figure 1* panels. (E–G) At P23 (late maturation phase), *Angpt4*<sup>+/Cre</sup>; *Rosa26*<sup>mT/mG</sup> retina shows GFP-positive cells enriched in the peripheral segment when compared to the central retina. Magnifications (F) and (G) reveal a GFP-positive cell population (arrowheads) connected to veins and perivenous capillaries.

DOI: https://doi.org/10.7554/eLife.37776.002

The following figure supplements are available for figure 1:

*Figure 1 continued on next page*

*Figure 1 continued*

**Figure supplement 1.** Generated *Angpt4* mouse alleles.
DOI: https://doi.org/10.7554/eLife.37776.003
**Figure supplement 2.** Controls of *Angpt4* gene expression in mouse retina model.
DOI: https://doi.org/10.7554/eLife.37776.004

ischemia (OIR model, *Figure 2—figure supplement 1A–E*) that both increased the number of Angpt4[+] astrocytes. By using conditional genetic depletion, we further confirmed that Angpt4 expression is positively regulated by HIF prolyl-hydroxylase 2 (Egln1) -dependent hypoxia pathway (*Figure 2H*). In hyperoxia, which in short-term promotes astrocyte differentiation (*Duan et al., 2017*), in long-term causes retinal degeneration (*Yamada et al., 1999*) and paradoxically can cause hypoxia by blocking vascular development in the retina (*West et al., 2005*), *Angpt1* mRNA expression was decreased while *Angpt2* and *Angpt4* mRNA levels increased. In addition, there was a trend for increased number of Angpt4[+] astrocytes (*Figure 2—figure supplement 2A–C*).

To investigate in detail when Angpt4 expression initiates and to compare Angpt4 expression pattern to Angpt1, Angpt2 and Tie2 especially in the peripheral retina, we performed in situ hybridization of antisense RNA probes on postnatal retina whole mounts (*Figure 3A–E*). At P2, one retina out of four showed a few Angpt4 expressing cells. At P3, *Angpt4* in situ signal was found consistently and it preceded the growing vascular front (*Figure 3A*). At P12, *Angpt4* mRNA was enriched in the astrocytes in the peripheral retina (*Figure 3B*), while no specific staining was detected in *Angpt4[-/-]* retina (*Figure 3—figure supplement 1A*). In situ hybridization thus confirmed the expression data obtained from *Angpt4[Cre]* and *Angpt4[LacZ]* models, but also revealed a few more Angpt4[+] cells than observed in the genetic models, possibly due to better sensitivity. In the peripheral retina, *Angpt1* mRNA was not detectable at P12 (*Figure 3C*), while *Angpt2* mRNA was found to be expressed in neuron cells in the intermediate layers of retina (*Figure 3D*) as reported earlier (*Hackett et al., 2000*). *Tie2* mRNA was localized in ECs in veins, arteries and capillaries (*Figure 3E*). Corresponding sense probes were used as controls (*Figure 3—figure supplement 1B–F*) and *Angpt1* antisense probe was validated by positive signal from P4 retina and P2 atriums (*Figure 3—figure supplement 1G–J*), as expected based on published expression data of Angpt1-GFP (*Kim et al., 2018*; *Park et al., 2017*).

## Angpt4 deficiency results in defective venous development in the peripheral retina

RT-qPCR confirmed the lack of *Angpt4* mRNA in *Angpt4[Cre/Cre]* mice (hereafter *Angpt4[-/-]*) and no compensatory change in *Angpt1* or *Angpt2* was detected (*Figure 1—figure supplement 1D*). As a first implication of a vascular phenotype, we observed weak alpha smooth muscle actin (αSMA) immunofluorescence in *Angpt4[-/-]* SMCs around the forming veins in the peripheral part of the retina at P12 (*Figure 4A–B*), correlating well with the spatiotemporal Angpt4 expression pattern. The low αSMA expression defect around veins persisted into adulthood (*Figure 4C–F*). Furthermore, the diameter of peripheral veins did not increase during development (*Figure 4J–K*). In both P12 and adult mice, the defective vascular morphogenesis was observed in the veins and preferentially in those locating at retina's periphery but not in central retina close to the optic nerve head (*Figure 4G–H,J–K*). In addition, arteries, peripheral arterioles and capillaries remained unaffected (*Figure 4C–F,I*) and no consistent alterations were observed in the vasculature of the intermediate or deep vascular plexuses postnatally or in adults (*Figure 4—figure supplement 1A–I*).

In WT mice, vein SMCs gradually acquire a matured phenotype characterized by increased cytoskeletal myofilaments and deposition of fibrillar collagen matrix evident in ultrastructural examination by transmission electron microscopy (TEM). Based on TEM, immature SMCs were present in the peripheral veins at P8 (*Figure 5A*). At later time points, however, ultrastructural analysis revealed lack of cellular and molecular characteristics of matured SMCs in *Angpt4[-/-]* mice (*Figure 5B–F,M–N*). To confirm defective maturation, whole mount retinas were stained for SM22, a marker of adult SMCs (*Li et al., 1996*), which was greatly reduced around *Angpt4[-/-]* veins (*Figure 5G–J*). Together with diminished αSMA and ultrastructural findings, this data indicated that Angpt4 is needed for SMC maturation. In vitro, angiopoietin receptor TIE2 was expressed predominantly in human ECs, but also in SMCs isolated from adult human veins or arteries (*Figure 5K*). Excess of ANGPT4,

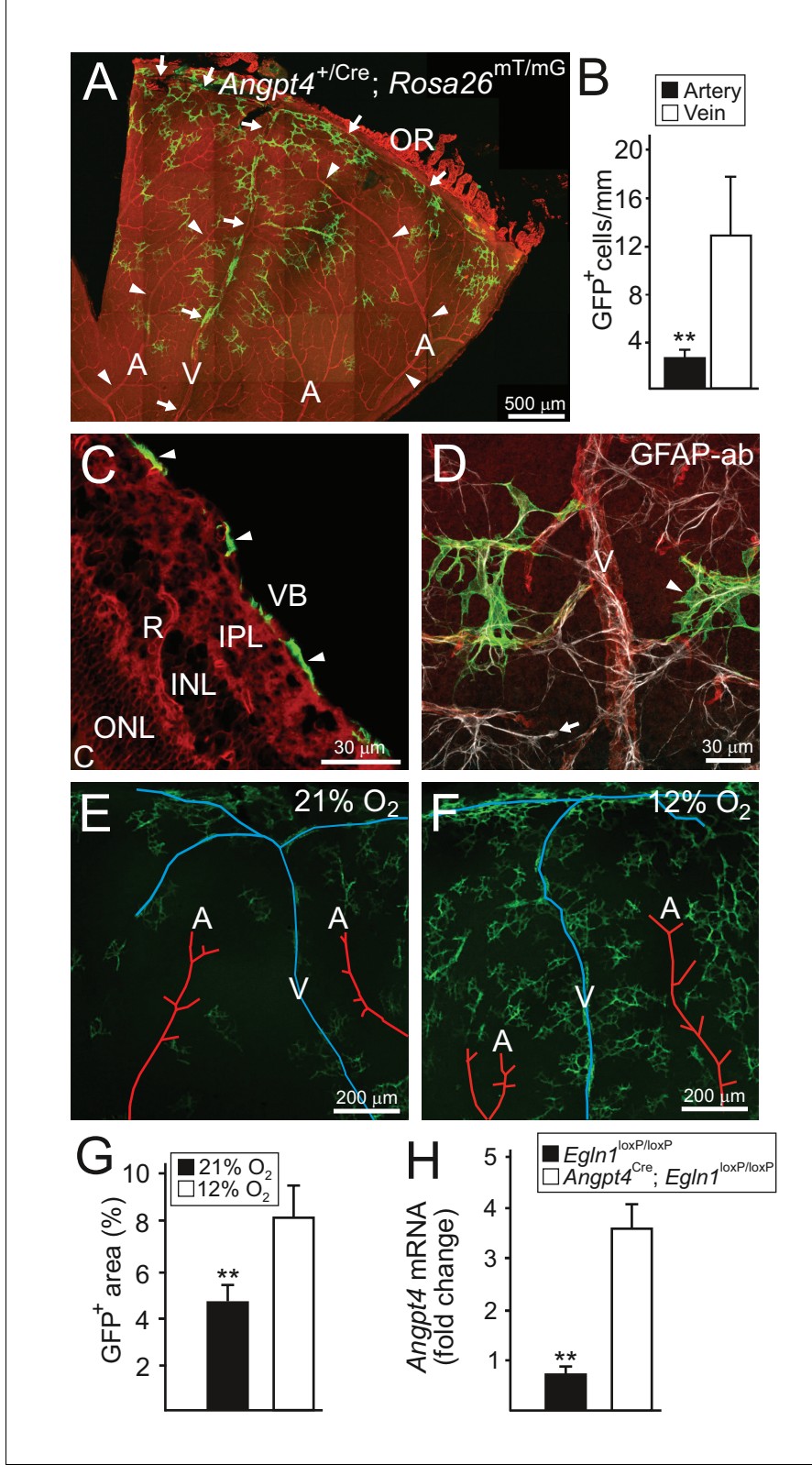

**Figure 2.** Angpt4 marks a specific subpopulation of retinal astrocytes and is induced in low oxygen. Angpt4 lineage cells (GFP, green) visualized in *Angpt4*^Cre/+; *Rosa26*^mT/mG reporter mice in adult (2 months) in (**A-D**) retina. Angpt4-negative cells express mTomato (red). (**A**) Similarly to developing retina (*Figure 1*), GFP-positive cells are enriched in the distal segment near ora serrata (OR) when compared to central retina. GFP positive cells are also

*Figure 2 continued on next page*

*Figure 2 continued*

less numbered near to arteries (A, arrowheads) than veins (V, arrows). Quantification of the image data is shown in (B). (C) Section through the adult retina (R) reveals Angpt4 lineage cells (arrowheads) locating in the superficial layer of retina, whereas other retina layers are negative (VB, vitreous body; IPL, inner plexiform layer; INL, inner nuclear layer; ONL, outer nuclear layer; C, choroid). (D) All GFP$^+$ cells (arrowhead) express astrocyte marker GFAP (white). However, not all astrocytes are GFP positive (arrow). (E–G) Low oxygen increases number of Angpt4$^+$ astrocytes, but does not induce Angpt4 expression in other cell types (in 1.5 months old mice). Quantification of the image data is shown in (G). (H) Induction of hypoxia-inducible factor pathway in Angpt4-expressing cells via *Angpt4*$^{Cre}$ mediated *Egln1* deletion increases *Angpt4* expression in P12 eye. Mean ±SD, **p<0.01 in t-test. *n* = 5 (B) and *n* = 4 mice (G)/group. In (H) *n* = 5 *Egln1*$^{loxP/loxP}$ and *n* = 2 *Angpt4*$^{Cre}$; *Egln1*$^{loxP/loxP}$ mice.

DOI: https://doi.org/10.7554/eLife.37776.005

The following figure supplements are available for figure 2:

**Figure supplement 1.** Induction of Angpt4 expression and degree of vasculopathy in OIR model in *Angpt4*$^{-/-}$ mice.

DOI: https://doi.org/10.7554/eLife.37776.006

**Figure supplement 2.** Angpt4 expression in hyperoxia model.

DOI: https://doi.org/10.7554/eLife.37776.007

however, did not increase SMC survival in Caspase 3/7 ELISA and did not promote SMC migration in scratch assay or their proliferation in BrdU ELISA (*Figure 5—figure supplement 1A–C*). qPCR from retinas did not reveal statistically significant changes in factors previously implicated in venous development, except a decrease in *Tie2* (*Figure 5L*). Collectively, these experiments indicated that Angpt4 is required for SMC maturation, but not for SMC recruitment, proliferation or survival and that Angpt4 does not directly signal to SMCs.

## Angpt4 deficiency results in neuronal cell alterations in the inner nuclear layer of the retina

Instead of a single common disease mechanism, neuronal edema is a complicated condition that results from multiple and still incompletely understood defects in fluid homeostasis caused by excessive fluid entry and/or insufficient clearance. In the retina, the transudate fluid is proposed to be absorbed back either into venous circulation, vitreous or choroid (*Daruich et al., 2018*). A model, in which one route is specially defective, should be particularly informative to study the importance of a putative fluid drainage mechanism. To evaluate the respective role of venous drainage, we investigated possible histological and ultrastructural changes in *Angpt4*$^{-/-}$ mice. Based on a laser photocoagulation induced mouse model for vein occlusion, edema is located mainly in the inner nuclear layer (INL) of the retina (*Fuma et al., 2017*). Therefore, we focused our histological analysis to INL, which is separated from the superficial vascular layer only by ~50 µm in mouse. In light microscopy, the thickness of the INL was increased in adult *Angpt4*$^{-/-}$ retinas (*Figure 6A–B*). TEM revealed enlarged, cell-organelle-free cytoplasm in neuronal cell somas in adult INL (*Figure 6D*), a phenotype that closely resembled ultrastructural changes of swollen cells (*Łotowska et al., 2009*). In addition, the normal architecture of Müller cells that have a specialized function in retinal fluid management expressing specific channels for water efflux (*Spaide, 2016*) was lost around the swollen cell bodies. These changes were not seen in P13 mice (*Figure 6C*). In adult mice, however, we found no alterations in the expression of AQP4 water (1.0 ± 0.2 vs.1.9 ± 1.5) or Kir4.1 potassium channels (1.0 ± 0.1 vs. 1.1 ± 0.3) (WT vs. *Angpt4*$^{-/-}$, mean ±SD, WT value set to 1, *n* = 3 in both genotypes, p=NS in t-test). Comparison of P13 and adult mice revealed that changes in neuronal cells occurred after defective venous development (*Figure 6C–D*) and were not found in other retinal layers than in INL (*Figure 6—figure supplement 1A–D*). We observed no evidence of cystoid edema or inflammation (lack of fluid-filled cysts and cellular infiltration in histology in *Figure 6A–B* and in ultrastructure in *Figure 6—figure supplement 1A–D*, and no increase in general inflammation markers in *Figure 6—figure supplement 2A–B*).

## Decreased venous drainage and neuronal activity in *Angpt4*$^{-/-}$ retinas

We next investigated possible functional changes in venous flow and neuronal activity. Fluorescein angiography (FA) was used to investigate blood circulation and vascular leakage, and flash

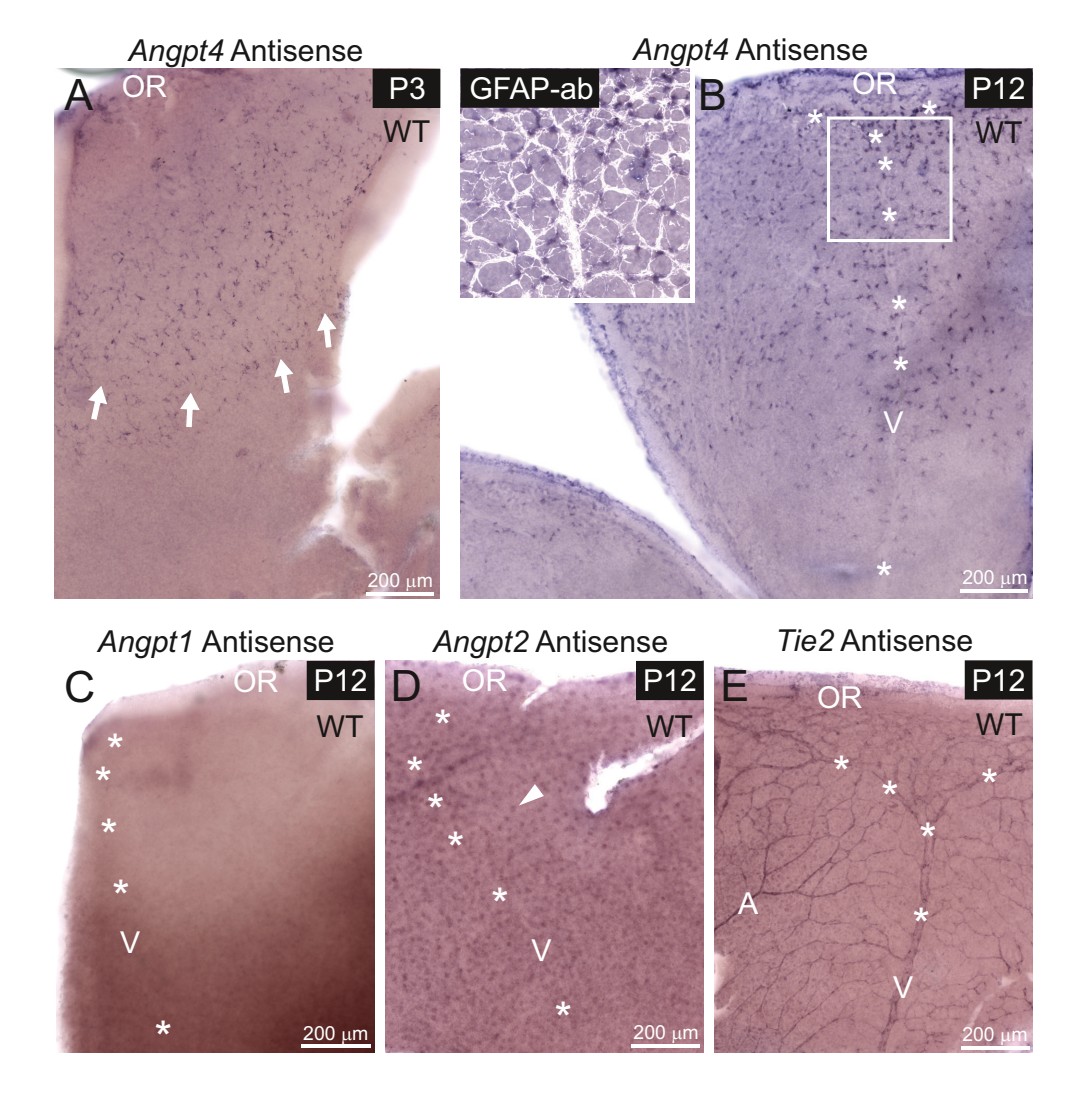

**Figure 3.** mRNA expression of angiopoietins and *Tie2* in retinal vein development. mRNA expression was detected in whole mount retinas from WT mice by in situ hybridization. (**A**) At P3, astrocytes expressing *Angpt4* precede the front of the developing vasculature (red blood cells indicated by arrows). (**B**) At P12, *Angpt4* expression is notable around the developing vein in the peripheral retina (V, asterisks) and colocalizes with the astrocyte marker GFAP (white overlay in insert). (**C**) *Angpt1* expression is not detected at P12 by whole mount in situ hybridization in the peripheral retina. (**D**) *Angpt2* is expressed in retinal neurons (arrowhead) in the intermediate retinal plexus. (**E**) Expression of *Tie2* is detected in the endothelium of arteries (A), veins and capillaries at P12. OR, ora serrata.

DOI: https://doi.org/10.7554/eLife.37776.008

The following figure supplement is available for figure 3:

**Figure supplement 1.** Controls of in situ hybridization staining.

DOI: https://doi.org/10.7554/eLife.37776.009

electroretinography (fERG) to evaluate neural retinal function. In FA, sodium fluorescein was injected intraperitoneally and the entering of fluorescent tracer to the retinal circulation was followed by video recording. In the early phase, venous filling in relation to arteries was decreased in *Angpt4*[-/-] mice (*Figure 7A,C,E*). No leakage was observed in the peak phase of fluorescence (*Figure 7B,D*). Limitation in fundus camera imaging is that the far peripheral retina is not fully visible; therefore, in addition to FA, we further investigated venous function in the peripheral location in whole-mount preparations after injection of fluorescent spheres into carotid artery. In *Angpt4*[-/-] mice, retention of fluorospheres was evident in the peripheral annular vein bifurcation sites, suggesting slower draining time than in controls (*Figure 7F–H*). In aged (eight mo) *Angpt4*[-/-] mice, fERG demonstrated a

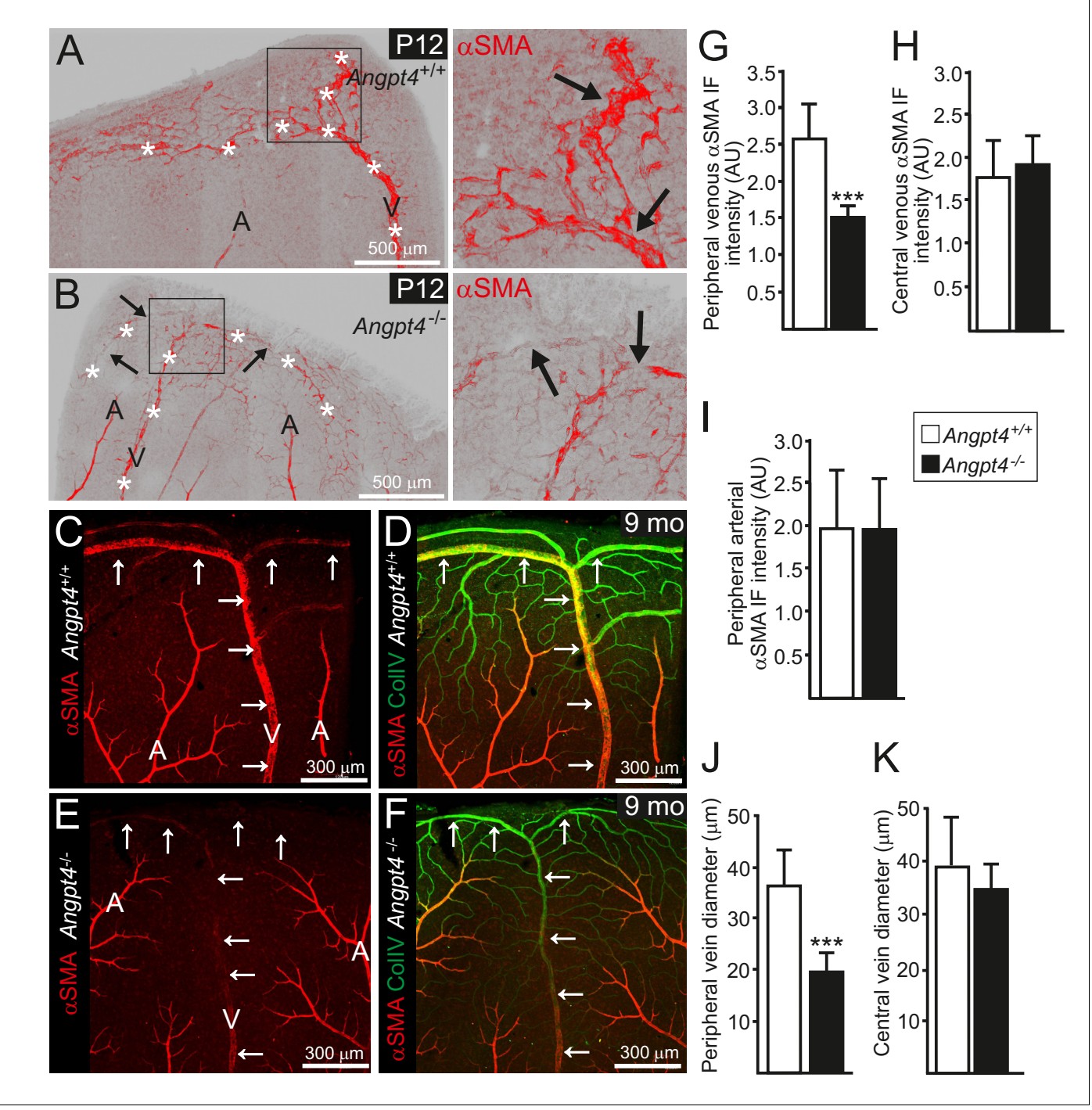

**Figure 4.** Retinal veins of Angpt4 deficient mice show SMC abnormality and smaller diameter. (**A**) P12 WT retina whole mount shows strong αSMA staining (red, arrows) in the developing annular vein in the peripheral retina. (**B**) αSMA staining is weak in *Angpt4*−/− retina especially in the peripheral location (red, arrows). Asterisks indicate developing Y-shaped annular peripheral vein in (**A**) and (**B**). (**C–F**) Whole mount retinal staining of nine months old mice. αSMA staining (red) (**C, E**) and merged image (**D, F**) with Col IV (green) indicate perivascular sleeves. In *Angpt4*−/− retina, αSMA level is low around annular veins (V, arrows) in the peripheral retina, but not around arteries (A). (**G–I**) αSMA immunofluorescence intensity (IF) in *Angpt4*−/− retina is reduced around annular peripheral veins (**G**), but not in veins close to optic nerve head (central) (**H**) or arteries (**I**) (n = 3 mice/group). (**J–K**) Peripheral annular vein diameter is reduced in *Angpt4*−/− retina. n = 8 WT and 6 *Angpt4*−/− mice, measured from the base of the peripheral bifurcation (**J**). n = 8 WT and 8 *Angpt4*−/− mice, measured from the central region close the optic nerve head (**K**). Mean ±SD, ***p<0.001 in t-test.
DOI: https://doi.org/10.7554/eLife.37776.010

The following figure supplement is available for figure 4:

*Figure 4 continued*

**Figure supplement 1.** Vascular structures and densities in the intermediate and deep vascular plexuses.
DOI: https://doi.org/10.7554/eLife.37776.011

decrease in b-wave amplitude that measures the activity of the bipolar, Müller and amacrine cells (*Figure 7I*). In contrast, a-wave amplitude, generated by photoreceptors, was not affected (*Figure 7—source data 1*). We found no evidence for vascular leakage in fluorescein angiography (*Figure 7D*) or hypoxia (negative pimonidazole staining and mRNA markers in *Figure 7—figure supplement 1A–D*).

## Angpt4 and ANGPT4 expand endothelial lumen structure via the TIE2/PI3K/serine protease pathway

In addition to defective SMC maturation, luminal diameter was smaller in *Angpt4^{-/-}* veins measured from retina whole mounts (*Figure 4J–K*) and TEM sections (*Figure 5O*), indicating halted venous development. To investigate molecular pathways via which Angpt4 may induce expansion of the endothelial luminal structure, human umbilical vein endothelial cell (HUVEC) spheroids were stimulated with Angpt4 and ANGPT4 in the fibrin gel model. Using inhibitors against plasmin/serine proteases (aprotinin), PI3K (BYL719 and LY294002) and shRNA constructs directed against TIE2 (*Figure 8A–D*), we found that similarly to ANGPT1, Angpt4 and ANGPT4 induced growth of luminal structures TIE2, PI3K and serine protease dependently. In a co-culture spheroid model, ANGPT4-induced lumen enlargement was not enhanced by SMCs (*Figure 8—figure supplement 1A*) and SMC spheroids alone did not respond to ANGPT4 (*Figure 8—figure supplement 1B*).

## Angpt4 and ANGPT4 show both ligand-specific and redundant functions among the angiopoietin family

The observed phenotype in *Angpt4^{-/-}* retinas was clearly different from that observed in mice lacking Angpt1 (*Lee et al., 2013*) or Angpt2 (*Gale et al., 2002*; *Hackett et al., 2002*) that both result in reduced growth of primary capillary plexus and not in vein-specific phenotype. In addition, in OIR model, *Angpt1* (*Lee et al., 2013*) and *Angpt2* (*Hackett et al., 2002*) deletions reduce neovascular tuft formation, while Angpt4 deficiency has no such effect (*Figure 2—figure supplement 1A–E*). To investigate if different vascular phenotypes may reflect differential molecular and cellular characteristics between the ligands, we next compared angiopoietins in in vitro assays (*Figure 9A–H*). ANGPT1 induces TIE2 translocation and phosphorylation in EC–EC junctions to activate Akt signaling in confluent ECs that increases endothelial stability (*Fukuhara et al., 2008*; *Saharinen et al., 2008*). Similarly to ANGPT1, but opposite to ANGPT2, Angpt4 and ANGPT4 induced TIE2 phosphorylation in inter-endothelial junctions (*Figure 9A–B*). In sparse migrating (*Fukuhara et al., 2008*; *Saharinen et al., 2008*) and spreading ECs (*Pietilä et al., 2012*), ANGPT1 and ANGPT2 induce TIE2 translocation to specific ECM contact sites to regulate EC adhesion and migration. In contrast to ANGPT1 and ANGPT2, Angpt4 and ANGPT4 did not bind to EC ECM (*Figure 9D*). After prolonged stimulation, the majority of ANGPT1-bound TIE2 is translocated into retraction fibers that represent former EC-ECM adhesions. Interestingly, ANGPT2/TIE2 was mainly located in the cell rear and not incorporated into permanent EC-ECM contacts, while ANGPT4 did not induce TIE2 translocation to ECM contacts to a significant extent (*Figure 9C*). When substrate-bound by solid phase antibody, Angpt4 and ANGPT4 failed to promote EC spreading that was induced by substrate-bound ANGPT1 (*Figure 9E–F*). To test if differences in adhesion are related to differential binding characteristics to TIE2, angiopoietins were compared in ELISA and surface plasmon resonance (SPR) assays (*Figure 9G–H*). In both assays, Angpt4 and ANGPT4 showed lower affinity to TIE2 than ANGPT1 and ANGPT2. We further evaluated the diffusion and kinetics of the ANGPT/TIE2-GFP complexes by the fluorescence recovery after photobleaching (FRAP) method (*Figure 9—figure supplement 1A–C*). Among the ligands, recovery rate was highest in Angpt4- and ANGPT4-stimulated cells, whereas ANGPT1 stimulation resulted in slowest recovery. Collectively, in vitro characterization indicated that similarly to ANGPT1, Angpt4 and ANGPT4 activated TIE2 in EC–EC junctions. When compared to other angiopoietins, binding to TIE2 and EC ECM was weaker, likely resulting in less stable ligand/TIE2 complexes evident in FRAP and prolonged stimulation experiments. To further compare

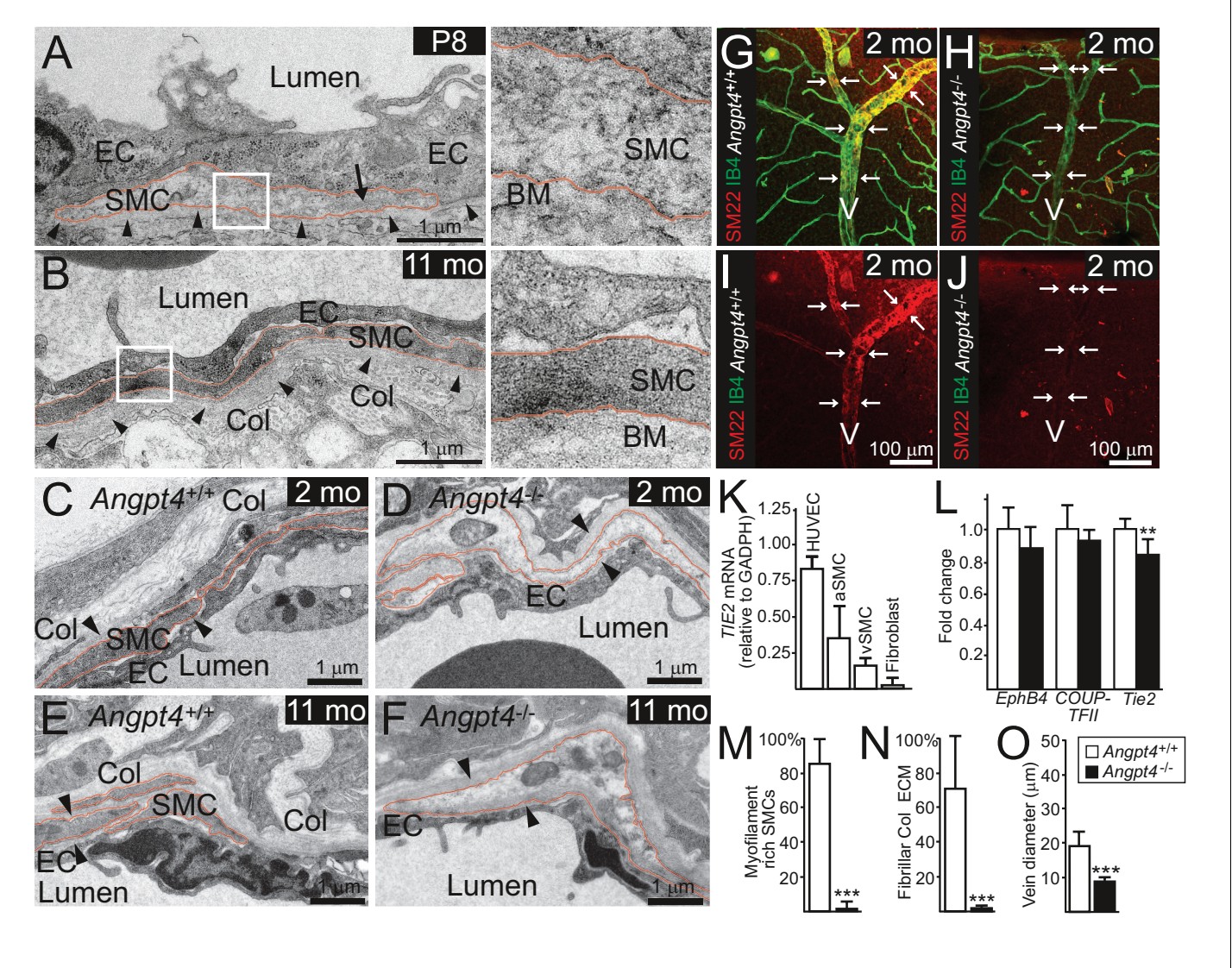

**Figure 5.** Angpt4 deficiency results in defective SMC maturation in veins. TEM micrographs of peripheral annular vein in P8 (A) and adult (11 months old) mice (B). SMCs are marked by red line. In the developing veins, SMCs are covered by basement membranes (BM, arrowheads) and SMC cytoskeleton appears translucent with some myofilaments (arrow). Perivascular ECM contains no fibrillar collagen. In adult mouse, venous SMC cytoplasm is mostly filled with myofilaments and perivascular matrix contains fibrillar collagen (Col). TEM micrographs of peripheral vein in 2 (C, D) and 11 (E, F) months old WT and *Angpt4-/-* mice, respectively. In both genotypes, perivascular cells (red line) are closely aligned with ECs and surrounded by BM (arrowheads) that indicates SMC identity. In *Angpt4-/-* retina, SMC cytoplasm lacks myofibril-rich cytoskeleton and fibrillar collagen (Col) matrix. No ultrastructural abnormalities are observed in ECs. (G–J) Whole mount retinal staining of peripheral veins in 2 months old WT (G, I) and *Angpt4-/-* (H, J) mice. SM22 (red) (G–J) and IB4 (green) (G, H) staining. In *Angpt4-/-* retina, SM22 level is low around veins (arrows, V). (K) Tie2 expression in arterial (a) and venous (v) SMC cell lines. HUVECs and fibroblasts are positive and negative controls, respectively. (L) qPCR analysis of venous specification marker genes at P12 retinas (*Angpt4-/-* n = 5 and WT n = 6). (M–N) Quantification of SMC maturation markers from TEM analyzed retinas (n = 6 WT and n = 7 *Angpt4-/-* mice). (O) Peripheral annular vein diameter is reduced in *Angpt4-/-* retina (n = 7 mice/group, the shortest diameter of peripheral branch measured from TEM cross-section). Data are presented as mean ±SD, ***p<0.001 and **p<0.01 *Angpt4-/-* vs. WT in t-test.

DOI: https://doi.org/10.7554/eLife.37776.012

The following figure supplement is available for figure 5:

**Figure supplement 1.** Smooth muscle cell proliferation, migration or apoptosis is not affected by ANGPT4.

DOI: https://doi.org/10.7554/eLife.37776.013

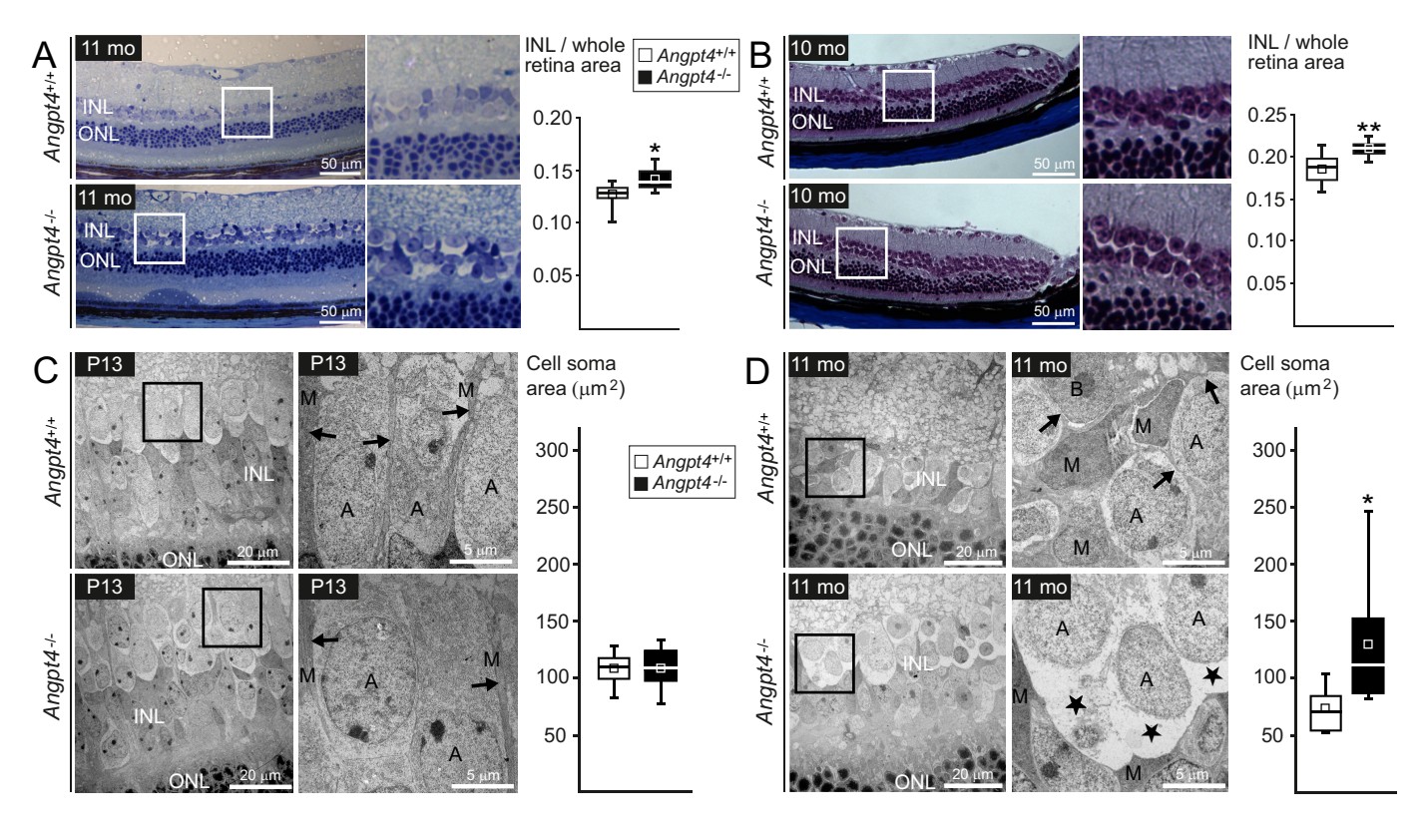

**Figure 6.** Neuronal cell swelling in adult *Angpt4*[-/-] mice. (A–B) Light microscopy images of peripheral retinal sections from control and *Angpt4*[-/-] mice stained with (A) toluidine blue (*n* = 7 mice/group) or (B) Masson's trichrome (*n* = 9 control and 5 *Angpt4*[-/-] mice). Framed area of inner nuclear layer (INL) is magnified on the right. Two different fixation, embedding and staining protocols indicate increased thickness of the INL in (A) and (B), data analyzed at least from four sections per mouse. (C–D) TEM micrographs from peripheral INL. (C) *n* = 8 control and 6 *Angpt4*[-/-] P13 mice and in adult mice (D) *n* = 8 control and 8 adult *Angpt4*[-/-] mice. Quantification (on the right) was done from cell somas locating in the interface between INL and inner plexiform layer that is composed mainly of amacrine cells (*Jeon et al., 1998*). Asterisks indicate cell swelling in (D) and black arrows Müller cell processes (C, D) that are displaced around swollen cells. A, amacrine cell; M, Müller glia; B, bipolar cell. (A–D) Median (line), average (square), 75th quartile (box), 5th and 95th percentile (whiskers) in whisker blots. *p<0.05 and **p<0.01 WT vs. *Angpt4*[-/-] in t-test.

DOI: https://doi.org/10.7554/eLife.37776.014

The following source data and figure supplements are available for figure 6:

**Source data 1.** Cell soma areas of individual neuronal cells in the interphase between INL and IPL.

DOI: https://doi.org/10.7554/eLife.37776.017

**Figure supplement 1.** Ultrastructural analysis of retinal layers in adult mice.

DOI: https://doi.org/10.7554/eLife.37776.015

**Figure supplement 2.** Inflammatory markers are not increased in adult Angpt4-deficient retinas.

DOI: https://doi.org/10.7554/eLife.37776.016

biochemical characteristics of angiopoietins used in this study, purified recombinant proteins were analyzed by SDS polyacrylamide gel electrophoresis and their potency to induce TIE2 tyrosine phosphorylation (activation) was quantified by immunoprecipitation and western blotting (*Figure 9—figure supplement 2A–D*). As previously published (*Lee et al., 2004*; *Valenzuela et al., 1999*), ANGPT4 induced TIE2 tyrosine phosphorylation in HUVECs. In comparison, Angpt4 was less oligomerized (*Figure 9—figure supplement 2C–D*) and induced relatively low TIE2 activation in human ECs when total TIE2 proteins were analyzed in western blot (*Figure 9—figure supplement 2A–B*). This was consistent with immunostaining (*Figure 9A*) of Angpt4-stimulated HUVECs, where TIE2 activation was observed as punctuate staining specially locating in EC-EC junctions, while a vast majority of TIE2 was evenly located on plasma membrane and was not stained with antibody specific for phosphorylated TIE2-Tyr992 on the kinase activation loop (*Nätynki et al., 2015*).

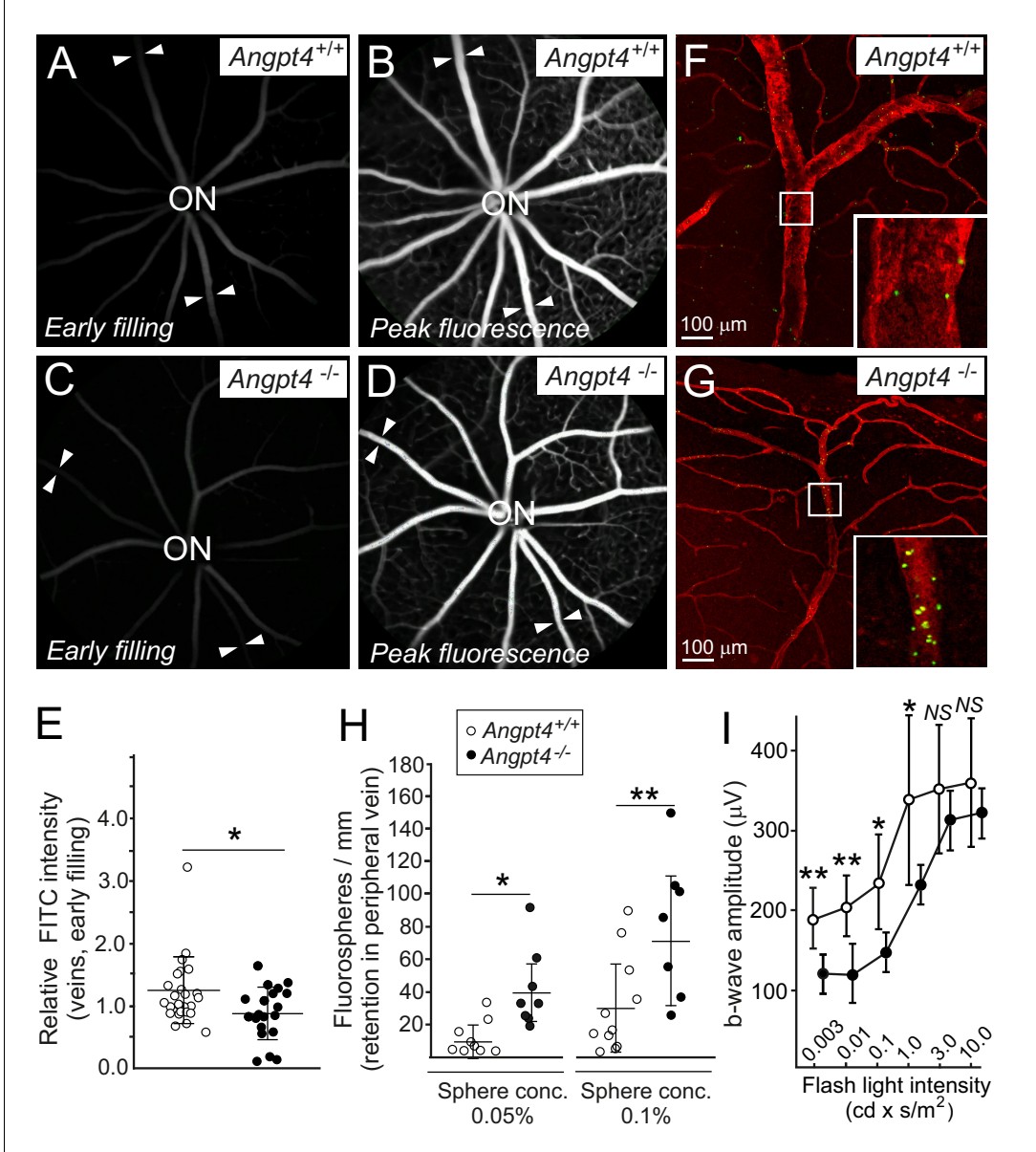

**Figure 7.** Functional analysis of venous flow and neuronal activity in retina. (A-D) Filling of the retinal veins in fluorescein angiography. Fundus camera images show the same retinas in (A) and (B) and in (C) and (D) in early filling phase (A, C) and in the peak phase of fluorescence (B, D). Arrowheads highlight examples of veins extending peripheral retina. (E) Quantification of the early filling image data; symbols represent fluorescein intensity in individual veins/average arterial fluorescein intensity in the same eye from 7 WT and 5 *Angpt4*-deficient mice. (F–G) Retention of fluorospheres in the peripheral annular vein after carotid artery injection. Blood vessels were counterstained with Cy3-conjugated αSMA antibody and co-injected Evans blue. (H) Two different sphere concentrations were used, and results are expressed as number of microspheres normalized by the length of venous segment analyzed. Each symbol represents an individual vein from 5 WT and 5 *Angpt4*-deficient mice. (I) Amplitudes of the b-wave in aged (eight months) WT and *Angpt4* deficient mice. ERGs were measured at six different flash intensities as indicated. Data are presented as mean ±SD ($n$ = 12 WT and $n$ = 8 *Angpt4$^{-/-}$* eyes). **$p<0.01$, *$p<0.05$ in t-test.

DOI: https://doi.org/10.7554/eLife.37776.018

The following source data and figure supplement are available for figure 7:

**Source data 1.** Electroretinogram measurements of individual eyes.
DOI: https://doi.org/10.7554/eLife.37776.020

**Figure supplement 1.** Hypoxia markers are not increased in adult Angpt4-deficient retinas.
DOI: https://doi.org/10.7554/eLife.37776.019

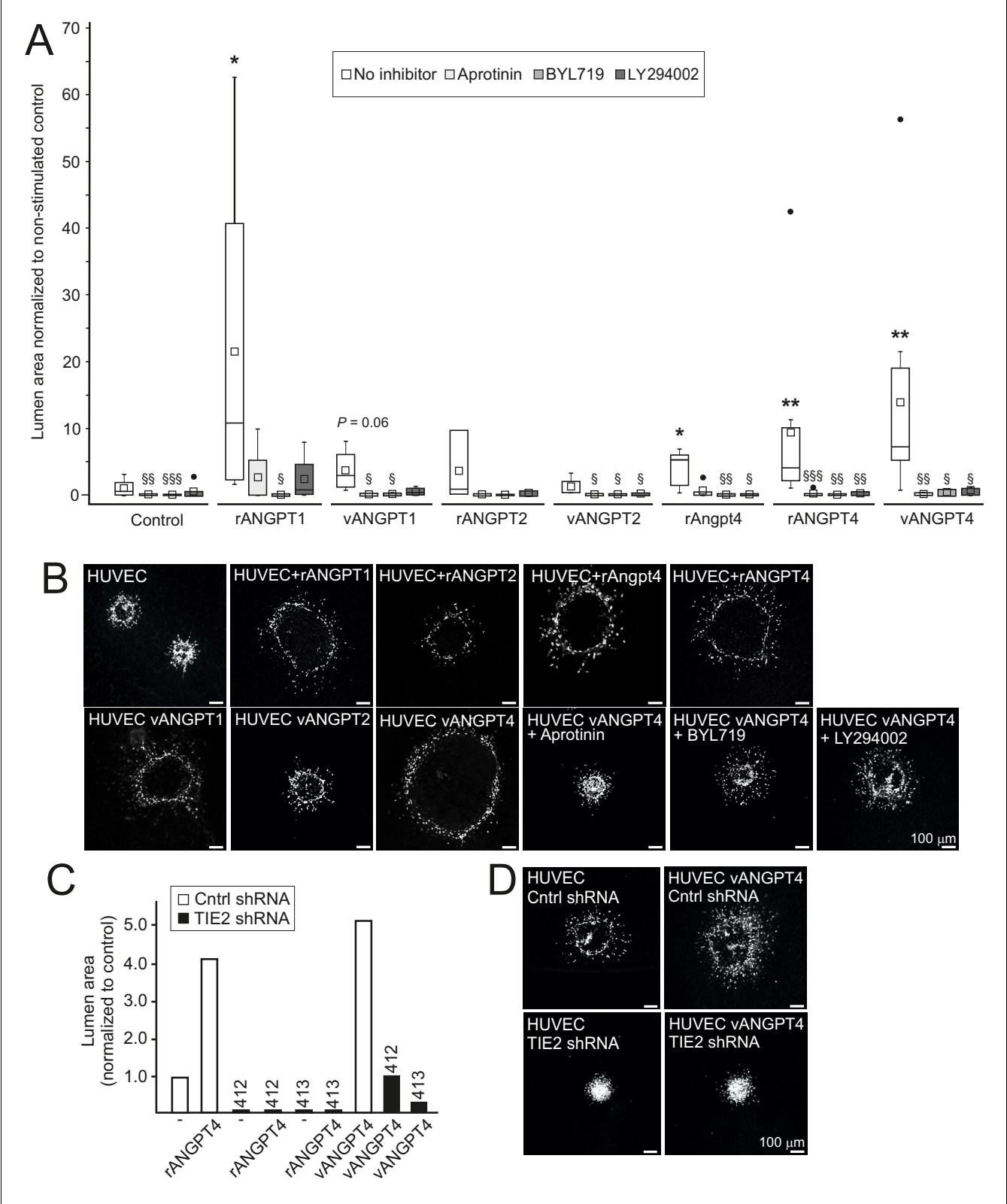

**Figure 8.** Angpt4 and ANGPT4 expand endothelial lumen structure via TIE2/PI3K/serine protease pathway. In vitro studies were performed by using mouse Angpt4 and human ANGPT4 purified recombinant proteins. (**A**) Lumen formation in HUVEC spheroids in fibrin gel angiogenesis model. Spheroids were stimulated with indicated recombinant (r) or virally produced (v) angiopoietins in the presence or absence of inhibitors indicated. Median (line), average (square), 75th quartile (box), 5th and 95th percentile (whiskers), outliers (•). **p<0.01, *p<0.05 stimulation vs. control, §§§p<0.001,

*Figure 8 continued on next page*

*Figure 8 continued*

§§p<0.01 and §p<0.05 inhibitor vs. no inhibitor in Kruskal–Wallis non-parametric ANOVA followed by Mann–Whitney U-test, *n* = 3 to 12 experiments. (B) Representative images of spheroids stimulated with angiopoietins and in the presence of inhibitors. (C) Lumen area in control or lentiviral TIE2 shRNA construct (412 or 413) treated HUVEC spheroids. Area of control shRNA lumen was set to 1. (D) Representative images of shRNA-treated spheroids.
DOI: https://doi.org/10.7554/eLife.37776.021
The following figure supplement is available for figure 8:

**Figure supplement 1.** Smooth muscle cells do not enhance responsiveness to ANGPT4 in EC co-cultures nor respond directly to ANGPT4.
DOI: https://doi.org/10.7554/eLife.37776.022

## Angpt4 expression outside of retina

As the neural retina can be considered as a part of central nervous system, we were interested in whether Angpt4 expressing astrocytes are also present in the brains. To support this suggestion, in a previous study ANGPT4 protein was reported to be expressed in normal, unselected population of human astrocytes at low level and increased in human glioma (*Brunckhorst et al., 2010*). However, in primary astrocyte cultures from WT mice (*Figure 10—figure supplement 1A*) or in *Angpt4*^Cre; *Rosa26*^mTmG brains (data not shown) we found no Angpt4 expression in astrocytes. To explore Angpt4 expression outside of the vasculature of the central nervous system, we analyzed mesenteric whole-mounts, where vasculature is readily visible for microscopic analysis. Interestingly, Angpt4 expression was found in arterial SMCs (*Figure 10*). To investigate the presence of circulating Angpt4, serum samples and the non-cellular supernatant of the eye that contained retinal extracellular soluble proteins and vitreous constituents were collected from WT and *Angpt4* deficient mice at P12. ELISA did not detect Angpt4 in mouse serum and quantity of soluble Angpt4 was low in the eye preparations indicating that there is no significant amount of circulating Angpt4 (*Figure 10—figure supplement 1B*).

## Discussion

In this study, we show that Angpt4 is necessary for venous remodeling and SMC maturation in the mouse retina. Lack of known growth factor(s) to initiate venous specification has hindered understanding the determination of venous fate, and therefore it is often considered as a default. Vascular SMCs are heterogenic cells that arise from multiple different origins in development (*Wang et al., 2015*). In the retina, vein SMCs can develop from neural crest origin (*Hughes and Chan-Ling, 2004*; *Trost et al., 2013*); however, thus far the molecular pathway(s) needed to initiate maturation of venous SMCs has been unknown. Our study identifies Angpt4 as the first growth factor having a venous-specific function in the mouse retina, thus providing new insights into poorly characterized developmental processes. From a translational perspective, identification of new molecules that regulate the formation of functional vasculature in the eye is necessary for the development of novel therapies that can have a major impact on the treatment of patients with limited treatment options. More broadly, understanding the mechanisms to establish an efficient circulatory loop involving functional veins is also crucial for development of regenerative medicine in general. Main findings and the mechanisms involved in the phenotypes observed in this study are presented in a graphical summary (*Figure 11*).

Mouse retinal vasculature develops postnatally in a highly organized and reproducible pattern. Formation of astrocyte network in the superficial retinal layer occurs between embryonic day 19 and first postnatal day thus preceding vascular development. Astrocytes have a special function to form a template for sprouting ECs, and together with retinal ganglion cells astrocytes express growth factors for angiogenic guidance and EC survival (*Selvam et al., 2018*). Interestingly, astrocytes are known to be a heterogeneous cell population (*Bayraktar et al., 2015*) and our data indicates that Angpt4+ astrocytes represent a specific subpopulation among the glial cells. Intriguingly, genetic deletion revealed that Angpt4 is especially important in the far peripheral segment that may develop via different pathway than central neural retina (*Venters et al., 2015*). In addition, clinical wide-field fundus imaging has identified a number of vascular diseases in humans in the peripheral retina that can affect also macula even though primary defect is peripheral (*Bajwa et al., 2015*; *Wessel et al.,*

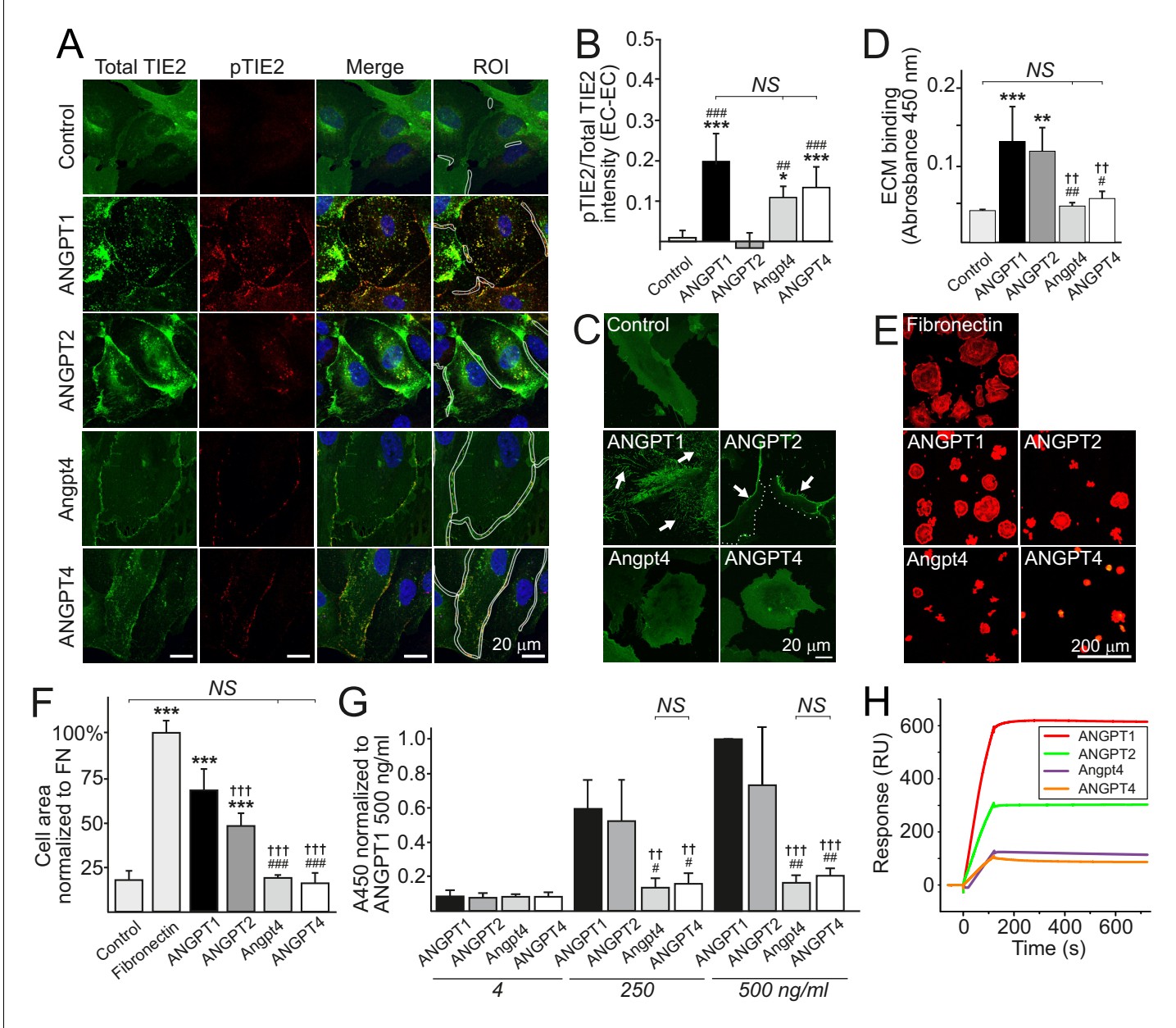

**Figure 9.** Ligand-specific and redundant functions of Angpt4 and ANGPT4. (**A**) Angiopoietin-induced TIE2 translocation and activation in cell–cell junctions. TIE2-WT HUVECs were left non-stimulated (control) or stimulated with recombinant angiopoietins as indicated for 1 hr, fixed, and stained for total TIE2 (green), phosphorylated TIE2 (pTIE2, red) and nuclei (DAPI, blue). White cropping indicates examples of ROIs (sites where cell contacts occurred based on microscopic examination) for TIE2 and pTIE2 intensity measurement from randomly selected cells. (**B**) Quantification of TIE2 activation (pTIE2/total TIE2) in cell–cell junctions ($n$ = 2 to 7 stimulations, total 330 to 915 cell junctions/stimulation were measured). (**C**) Sparse TIE2-GFP HUVECs were stimulated with angiopoietins overnight, fixed and imaged. In ANGPT1-stimulated cells, majority of TIE2-GFP is ECM bound (arrows). ANGPT2 promoted TIE2 translocation to the retracting cell edges (arrows) and less in ECM (dotted line indicates leading edge of the cell). Angpt4 and ANGPT4 did not induce long-term effect on TIE2 clustering and translocation into ECM. (**D**) Angiopoietin binding to acellularized ECM fraction from cultured HUVECs ($n$ = 4 run in triplicate). (**E**) TIE2-WT HUVECs spreading on fibronectin, ANGPT1, ANGPT2, Angpt4 or ANGPT4. Cells were let to spread on coated coverslips for 1 hr and stained for actin. Quantification of image data is shown in (**F**). Cell area was normalized to fibronectin (FN) ($n$ = 3 to 9 experiments, 333 to 1778 cells per coating were analyzed). (**G**) Angiopoietin-TIE2 binding affinity measured by ELISA. ANGPT1 binding to TIE2 (250 ng/ml) was set to one at ANGPT1 concentration 500 ng/ml ($n$ = 4 experiments). (**H**) Angiopoietin binding to TIE2 in surface plasmon resonance assay. Human purified TIE2-Fc was immobilized on CM5 chip, and angiopoietins were injected onto the TIE2-Fc surface at 100 nM concentration. Mean ±SD, ANOVA followed by the Bonferroni post hoc test. ***$p < 0.001$, **$p < 0.01$, *$p < 0.05$ vs. control; †††$p < 0.001$, ††$p < 0.01$ vs. ANGPT1; ###$p < 0.001$, ##$p < 0.01$, #$p < 0.05$ vs. ANGPT2.

DOI: https://doi.org/10.7554/eLife.37776.023

*Figure 9 continued on next page*

*Figure 9 continued*

The following figure supplements are available for figure 9:

**Figure supplement 1.** Fluorescence recovery of TIE2/angiopoietin complexes in FRAP analysis.
DOI: https://doi.org/10.7554/eLife.37776.024
**Figure supplement 2.** SDS polyacrylamide gel electrophoresis of recombinant purified angiopoietins used in this study.
DOI: https://doi.org/10.7554/eLife.37776.025

*2012*). Despite its emerged clinical importance, until now no mechanism specially regulating vascular remodeling in this domain has been described.

Angpt4 expression was not found in mouse brain astrocytes, which further strengthens our conclusion that Angpt4$^+$ astrocytes represent a specific subpopulation. Outside of retina, we found *Angpt4* expression in arterial SMCs. These results (negative brain astrocytes, expression in certain SMCs) are in line with a recent comprehensive single-cell transcriptomic study of the mouse brain vasculature (*Vanlandewijck et al., 2018*). Interestingly, in addition to retinal neuronal cells, Angpt1-GFP is also reported to be expressed in perivascular mural cells (*Park et al., 2017*). The data is currently limited but suggests that both Angpt1 and Angpt4 can be expressed in certain vascular SMCs, raising interesting questions regarding ligand-specific and possible synergistic or additional effects of these Tie2 ligands. Binding affinity of purified recombinant ANGPT4 protein to TIE2-Fc was considerably low when compared to ANGPT1 that binds TIE2 with high affinity. Nevertheless, ANGPT4 to TIE2 binding was sufficient for TIE2 phosphorylation (activation) in cell culture assays. In the case both ligands are present, the higher affinity may favor ANGPT1/TIE2 binding in tissues. Originally, ANGPT4 was identified as a TIE2 agonistic ligand in HUVECs (*Lee et al., 2004*; *Valenzuela et al., 1999*). In the first reports, mouse Angpt4 did not significantly induce TIE2 phosphorylation in human ECs when analyzed in western blot that suggested a species-specific effect (*Lee et al., 2004*; *Valenzuela et al., 1999*). However, in mice, excess of both Angpt4 and ANGPT4 similarly induced angiogenic remodeling in corneal micropocket assay (*Lee et al., 2004*) and in the trachea (*Kim et al., 2007*), indicating that Angpt4 and ANGPT4 have similar in vivo functions. In our study, Angpt4 promoted lumen formation of human ECs with somewhat lower extent, but we observed no differences in Angpt4 and ANGPT4 binding characteristics to ECM, ECs or TIE2. Collectively, this suggests that despite relatively low average sequence homology (65% identity) between Angpt4 and ANGPT4, certain functional characteristics are still conserved.

Interestingly, observed spatiotemporally regulated expression of Angpt4 in astrocytes in the peripheral retina close to the developing veins significantly differed from other angiopoietins (*Gale et al., 2002*; *Hackett et al., 2000*; *Hackett et al., 2002*; *Park et al., 2017*). In addition, the venous-specific phenotype we characterize in *Angpt4$^{-/-}$* mice also differs from defects in *Angpt1$^{-/-}$* and *Angpt2$^{-/-}$* retinas; *Angpt1* deletion at P2–4 (analyzed at P5) caused reduced vascular density (*Lee et al., 2013*), and *Angpt2* deficiency decreased sprouting of the primary plexus and delayed regression of the embryonic hyaloid vasculature at P10 (*Gale et al., 2002*). Although Angpt4 expression was induced in OIR model for ischemic retinopathy, Angpt4 deficiency did not affect microvascular proliferation, as has been found in *Angpt1$^{-/-}$* (*Lee et al., 2013*) and *Angpt2$^{-/-}$* mice (*Hackett et al., 2002*), suggesting that Angpt4 is functionally different. On the other hand, in addition to ligand-specific effects, our in vitro characterization also revealed redundant functions between angiopoietins 1 and 4, suggesting that differential, spatiotemporally regulated expression provides an important mechanism for complementary and sequential roles of angiopoietins to establish retinal circulation system. Moreover, based on reported venous-specific effects of inducible Tie2 deletion in retina (*Chu et al., 2016*) and over-activating TIE2 mutations in venous malformations (*Kangas et al., 2018*), Tie2 signaling appears particularly important to venous development that may contribute to the vessel-type-specific phenotype in *Angpt4$^{-/-}$* mice.

In Angpt4-deficient mice, we observed defective circumferential growth of peripheral veins postnatally and changes in neuronal cell organization and function in adult mice. Based on Poiseuille's equation, vessel flow is proportional to its radius. In line with this suggestion, functional tests using fluorescent tracers showed decreased venous flow in *Angpt4$^{-/-}$* retinas and we propose that alterations in neural cells are secondary to venous remodeling defect. While ultrastructural changes in neural cell somas closely resembled intracellular edema (*Łotowska et al., 2009*), in Müller cells we

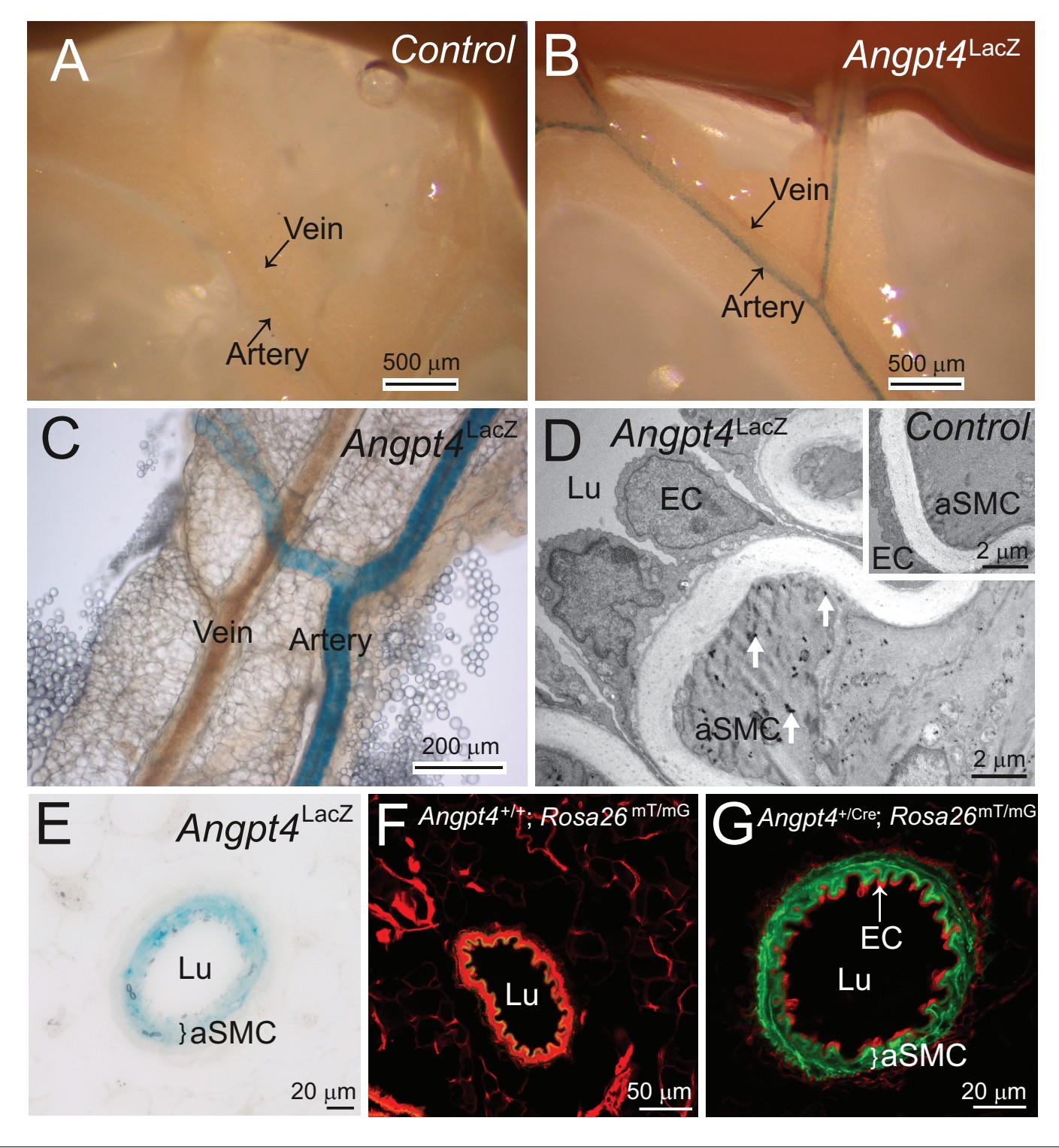

**Figure 10.** Outside of retina, *Angpt4* is expressed in muscular arteries in adult mice. (**A**) LacZ gene negative control. (**B–C**) Whole mount preparations of small intestine mesentery showed X-gal staining in *Angpt4*^LacZ^ mice arteries, but not in veins. (**D**) TEM analysis of *Angpt4*^LacZ^ mesentery artery showed X-gal staining as electron-dense precipitation (arrows) inside arterial smooth muscle cell (aSMC) in tunica media layer, whereas the ECs and SMCs in LacZ gene negative control (insert) did not contain precipitate. Lu, vessel lumen. (**E**) X-gal staining in tunica media layer (bracket) in the cross section of *Angpt4*^LacZ^ mesenteric artery. (**F**) Cross-section of mesenteric artery from negative control *Angpt4*^+/+^; *Rosa26*^mTmG^. Elastin shows green autofluorescence. (**G**) *Angpt4*^Cre^; *Rosa26*^mTmG^ fate mapping mouse line showed GFP signal in tunica media layer (bracket), whereas ECs were negative.
*Figure 10 continued on next page*

*Figure 10 continued*

DOI: https://doi.org/10.7554/eLife.37776.026

The following figure supplement is available for figure 10:

**Figure supplement 1.** Angpt4 is not present in serum or expressed in brain astrocytes.

DOI: https://doi.org/10.7554/eLife.37776.027

observed discontinuation of cellular processes in the affected areas but no increase in Müller cell volume in the INL. Changes in Müller cells occur in retinal pathologies involving edema and are associated with altered expression of AQP4 water or Kir4.1 potassium channel, that may result in dysfunction in water homeostasis in these cells (*Daruich et al., 2018*). In *Angpt4*[−/−] retinas, we found no changes in either AQP4 or Kir4.1, suggesting that water balance in Müller cells may be retained. Other neuronal cells in the INL (including amacrine and bipolar cells) may lack efficient water transport mechanism and thus be more vulnerable for alterations in fluid homeostasis due to diminished venous drainage that can be aggravated by the alterations in Müller cell architecture. We did not find evidence of interstitial edema, either. In literature, there is no similar pathology than what we observe in *Angpt4*[−/−] mice (congenital decrease in vein diameter, venous SMC abnormality, diminished vein flow, but no inflammation or leakage), making direct comparison to the better characterized retinal pathologies (such as diabetic retinopathy and retinal vein occlusion) difficult. It may be possible that interstitial edema is prominent in retinal vascular diseases in which vascular barrier is disrupted, which causes large increase in fluid entry that is excessive for exit mechanisms.

Overall, our study unraveled unknown physiological functions of poorly characterized Angpt4 in venous development and identified ligand-specific and redundant functions among the family of angiopoietins. Angpt/Tie signaling pathway has emerged as an important target for drug development in ophthalmological diseases (*Saharinen et al., 2017*). In addition to providing novel insights into developmental angiogenesis, thorough investigation of all angiopoietin ligands is necessary to fully evaluate the therapeutic potency of this signaling system.

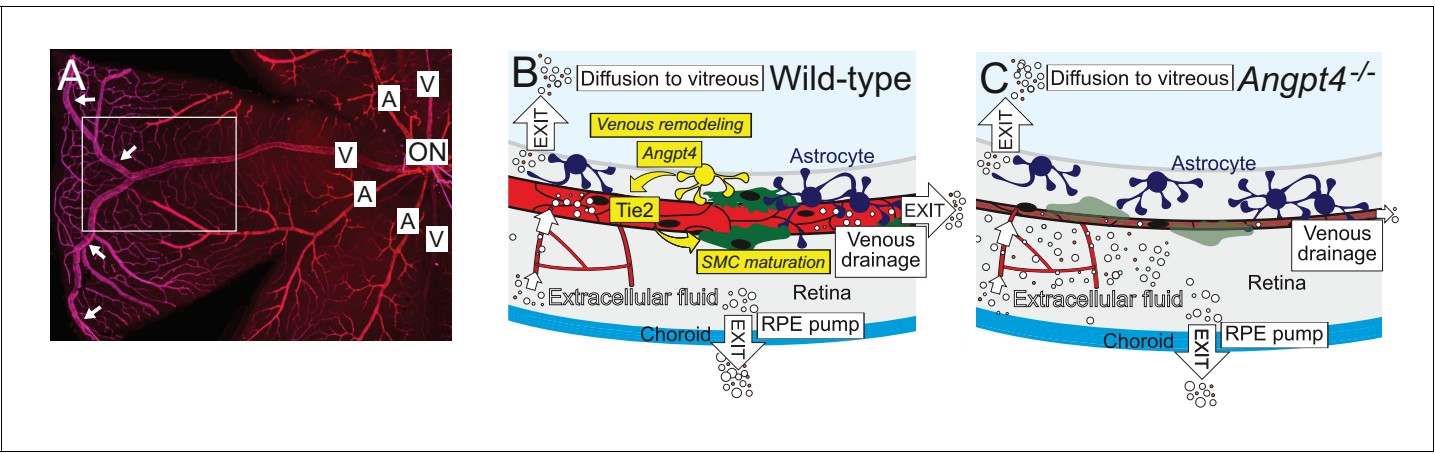

**Figure 11.** Graphical summary. (**A**) Whole mount staining of the superficial vascular plexus of adult mouse retina, one quarter is shown. In mouse, typically two large veins extend to the far periphery of the retina to form an annular vessel structure (arrows), where peripheral capillaries drain. A, artery; V, vein; ON, optic nerve head. Framed area is represented in (**B**) and (**C**). (**B**) Angpt4 is expressed in a specific subpopulation of hypoxia-regulated astrocytes that are enriched in the peripheral retina and locate close to the developing veins to promote venous remodeling and SMC maturation via endothelial cell-expressed Tie2. Potential fluid removal mechanisms from neural retina include retinal pigment epithelial (RPE) cell pumps, venous drainage and diffusion to vitreous (*Daruich et al., 2018*). (**C**) Angpt4 deficiency leads to defective circumferential growth and SMC maturation in early postnatal development that secondarily leads to neuronal cell damage due to impaired venous drainage. Tie2 signaling has a special importance in venous development (*Chu et al., 2016*; *Vikkula et al., 1996*), and in the peripheral retina, expression of competitive Angpt1 ligand is low that may contribute to the localized, vessel-type specific phenotype in Angpt4 deficient mice.

DOI: https://doi.org/10.7554/eLife.37776.028

# Materials and methods

## Key resources table

| Reagent type (species) or resource | Designation | Source or reference | Identifiers | Additional information |
|---|---|---|---|---|
| Strain, strain background (*Mus musculus*, C57BL/6N) | WT | Charles River Laboratories | | |
| Strain, strain background (*Mus musculus*, C57BL/6N) | *Angpt4*$^{LacZ}$ | This Paper | | Generation described in detail in Materials and methods |
| Strain, strain background (*Mus musculus*, C57BL/6N) | *Angpt4*$^{Cre}$ | This Paper | | Generation described in detail in Materials and methods |
| Strain, strain background (*Mus musculus*, C57BL/6N) | *Angpt4*$^{Cre}$; *Rosa26*$^{mTmG}$ | This Paper | | Generation described in detail in Materials and methods |
| Strain, strain background (*Mus musculus*, C57BL/6N) | *Egln1*$^{loxP}$ | Other | | A gift from Prof. Johanna Myllyharju, University of Oulu |
| Cell line (human) | Human umbilical vein endothelial cells (HUVEC) | Promocell | C-12203 | |
| Cell line (human) | TIE2-WT HUVEC | *Saharinen et al. (2008)* | | |
| Cell line (human) | TIE2-GFP HUVEC | *Saharinen et al. (2008)* | | |
| Cell line (human) | HUVEC vANGPT1 | This Paper | | Generation described in detail in Materials and methods |
| Cell line (human) | HUVEC vANGPT2 | This Paper | | Generation described in detail in Materials and methods |
| Cell line (human) | HUVEC vANGPT4 | This Paper | | Generation described in detail in Materials and methods |
| Cell line (human) | Human umbilical artery smooth muscle cells | Cell Applications | 252–05 n | |
| Cell line (human) | Human umbilical vein smooth muscle cells | Cell Applications | 250–05 n | |
| Cell line (human) | Human fibroblasts | ATCC | CCL-210; RRID:CVCL_2382 | |
| Antibody | αSMA, alpha smooth muscle cell, Cy3-conjugated (Mouse monoclonal) | Sigma-Aldrich | C6198; RRID:AB_476856 | IF (1:300) |
| Antibody | IB4, biotinylated isolectin B4 | Vector laboratories | B-1205; RRID:AB_2314661 | IF (1:300) |

*Continued on next page*

*Continued*

| Reagent type (species) or resource | Designation | Source or reference | Identifiers | Additional information |
|---|---|---|---|---|
| Antibody | ColIV (rabbit polyclonal) | Merck Millipore | AB756P; RRID:AB_2276457 | IF (1:300) |
| Antibody | GFAP (rabbit polyclonal) | Abcam | ab7260; RRID:AB_305808 | IF (1:300) |
| Antibody | Pimonidazole (rabbit antisera) | Hypoxyprobe | Pab2627; RRID:AB_1576904 | IF (1:300) |
| Antibody | SM22alpha (goat polyclonal) | Abcam | ab10135; RRID:AB_2255631 | IF (1:300) |
| Antibody | Tie2, clone Ab33 (mouse monoclonal) | Merck Millipore | 05–584; RRID:AB_309820 | IF (1:100); WB (1:1000) |
| Antibody | Tie2 (rabbit polyclonal) | Santa Cruz Biotechnology | sc-324; RRID:AB_631102 | WB (1:1000) |
| Antibody | pTie2 (Tyr992) (rabbit polyclonal) | Cell Signaling | 4221S; RRID:AB_2203198 | IF (1:100); WB (1:1000) |
| Antibody | phospho-tyrosine, clone 4G10 (mouse monoclonal) | Merck Millipore | 05–321; RRID:AB_309678 | WB (1:1000) |
| Antibody | penta-HIS (mouse monoclonal) | Qiagen | 34660; RRID:AB_2619735 | WB (1:1000) |
| Antibody | TRITC-phalloidin | Sigma-Aldrich | P1951; RRID:AB_2315148 | IF (1:2000) |
| Antibody | Alexa Fluor 488, streptavidin-conjugated | Jackson Immuno Research | 016-540-084; RRID:AB_2337249 | IF (1:300) |
| Antibody | Alexa Fluor 488, Donkey anti-rabbit | Jackson Immuno Research | 711-545-152; RRID:AB_2313584 | IF (1:300) |
| Antibody | Alexa Fluor 488, Donkey anti-mouse | Jackson Immuno Research | 715-545-151; RRID:AB_2341099 | IF (1:300) |
| Antibody | Alexa Fluor 647, Goat anti-rabbit | Jackson Immuno Research | 111-605-045; RRID:AB_2338075 | IF (1:300) |
| Antibody | Cy3, Donkey anti-goat | Jackson Immuno Research | 705-165-147; RRID:AB_2307351 | IF (1:300) |
| Antibody | Peroxidase-conjugated goat anti-mouse | Jackson Immuno Research | 115-035-003; RRID:AB_10015289 | WB (1:10000) |
| Antibody | Peroxidase-conjugated goat anti-rabbit | Jackson Immuno Research | 111-035-003; RRID:AB_2313567 | WB (1:10000) |
| Antibody | Anti-Digoxigenin-AP, Fab Fragments | Roche | 11093274910; RRID:AB_514497 | ISH (1:1500) |
| Antibody | Protein A/G PLUS-Agarose | Santa Cruz biotechnology | sc-2003; RRID:AB_10201400 | |
| Sequence-based reagent | qPCR primers | Sigma-Aldrich | | *Supplementary file 1* |

*Continued on next page*

*Continued*

| Reagent type (species) or resource | Designation | Source or reference | Identifiers | Additional information |
|---|---|---|---|---|
| Sequence-based reagent | mouse *Angpt1* in situ RNA Antisense probe | This Paper | | *Supplementary file 2*; Generation described in detail in Materials and methods |
| Sequence-based reagent | mouse *Angpt2* in situ RNA Antisense probe | This Paper | | *Supplementary file 2*; Generation described in detail in Materials and methods |
| Sequence-based reagent | mouse *Angpt4* in situ RNA Antisense probe | This Paper | | *Supplementary file 2*; Generation described in detail in Materials and methods |
| Sequence-based reagent | mouse *Tie2* in situ RNA Antisense probe | This Paper | | *Supplementary file 2*; Generation described in detail in Materials and methods |
| Sequence-based reagent | mouse *Angpt1* in situ RNA Sense probe | This Paper | | *Supplementary file 2*; Generation described in detail in Materials and methods |
| Sequence-based reagent | mouse *Angpt2* in situ RNA Sense probe | This Paper | | *Supplementary file 2*; Generation described in detail in Materials and methods |
| Sequence-based reagent | mouse *Angpt4* in situ RNA Sense probe | This Paper | | *Supplementary file 2*; Generation described in detail in Materials and methods |
| Sequence-based reagent | mouse *Tie2* in situ RNA Sense probe | This Paper | | *Supplementary file 2*; Generation described in detail in Materials and methods |
| Sequence-based reagent | Tie2-412 shRNA | The RNAi consortium | | http://www.broadinstitute.org/rnai/trc |
| Sequence-based reagent | Tie2-413 shRNA | The RNAi consortium | | http://www.broadinstitute.org/rnai/trc |
| Sequence-based reagent | PLKO.1 shRNA | The RNAi consortium | | http://www.broadinstitute.org/rnai/trc |
| Peptide, recombinant protein | ANGPT1 | R & D Systems | 923-AN-025 | |
| Peptide, recombinant protein | COMP-ANGPT1 | Other | | A gift from Prof. Gou Young Koh, KAIST |
| Peptide, recombinant protein | ANGPT2 | R & D Systems | 623-AN-025 | |
| Peptide, recombinant protein | Angpt4 | R & D Systems | 738-AN-025 | |
| Peptide, recombinant protein | ANGPT4 | R & D Systems | 964-AN-025 | |

*Continued on next page*

*Continued*

| Reagent type (species) or resource | Designation | Source or reference | Identifiers | Additional information |
|---|---|---|---|---|
| Commercial assay or kit | Mouse Angiopoietin-3 DuoSet ELISA | R & D Systems | DY738 | |
| Commercial assay or kit | RNeasy Mini Kit | Qiagen | 74104 | |
| Commercial assay or kit | Fibrous Tissue Mini Kit | Qiagen | 74704 | |
| Commercial assay or kit | Caspase-Glo 3/7 Assay | Promega | G8091 | |
| Commercial assay or kit | Cell proliferation ELISA, BrdU | Roche | 11647229001 | |
| Chemical compound, drug | pimonidazole | Hypoxyprobe | HP3-100Kit | |
| Chemical compound, drug | Evans blue | Sigma-Aldrich | E2129 | |
| Chemical compound, drug | Carboxylate-modified fluorescent beads | Invitrogen | F888 | |
| Chemical compound, drug | 5% fluorescein sodium salt | Sigma-Aldrich | F6377 | |
| Chemical compound, drug | Aprotinin | Sigma-Aldrich | A3428 | |
| Chemical compound, drug | BYL719 | LC Laboratories | A-4477 | |
| Chemical compound, drug | LY294002 | Selleck | S1105 | |
| Chemical compound, drug | Protease inhibitor | Sigma-Aldrich | P8340 | |
| Chemical compound, drug | Phosphatase inhibitor | Sigma-Aldrich | P5726 | |
| Chemical compound, drug | Tie2-Fc | R & D Systems | 313-TI | |
| Chemical compound, drug | NBT/BCIP stock solution | Roche | 11681451001 | |
| Software, algorithm | Olympus Fluoview Ver.3.0 Viewer | Olympus | RRID:SCR_014215 | |
| Software, algorithm | ZEN 2012 (blue edition) | Carl Zeiss | RRID:SCR_013672 | |
| Software, algorithm | Fiji (ImageJ) | Fiji | RRID:SCR_002285 | |
| Software, algorithm | Biacore T200 Evaluation Software, version 2.0 | Biacore | | |
| Software, algorithm | OriginPro 2018b | OriginLab Corporation | RRID:SCR_014212 | |
| Software, algorithm | Microsoft Excel 2016 | Microsoft Office | RRID:SCR_016137 | |
| Other | DAPI | Sigma-Aldrich | D9542 | IF (1:500) |

## Statistics

Kruskal–Wallis and Mann–Whitney *U* tests were used in *Figure 8A*. Otherwise, comparisons between two groups were done using two-tailed Student's t-test and multiple comparison with ANOVA followed by the Bonferroni post-hoc test.

## Study approval

Experiments involving mice were performed under permissions from the National Animal Experiment Board following the regulations of the EU and national legislation.

## Generation, maintenance and genotyping of the transgenic mouse lines

For *Angpt4*<sup>LacZ</sup> mouse model, clone HTGRS06013 A 1 C06 was obtained from the trans-NIH Knock-Out Mouse Project (KOMP) and the KOMP Repository (www.komp.org). The AsiSI-linearized targeting construct was electroporated into B6 ES cells and selected with G418. Genomic DNA isolated from resistant colonies was screened by PCR. The PCR assay for the targeted *Angpt4* gene used the primer pair 5′-cacaacgggttcttctgttagtcc-3′ from construct and 5′-gcgaaggagcggctgagcctgagaactc-3′ from *Angpt4* gene. Correctly targeted clones were injected into C57BL/6N blastocysts, and two separate mouse lines were generated by standard methods.

Targeting vector for *Angpt4*<sup>Cre</sup> mouse line was generated by using recombineering approach as described previously (*Wang et al., 2006*). Briefly, BAC-clones RP23-204D14 and RP23-131K21 overlapping genomic *Angpt4* locus were obtained from BACPAC resources (Children's Hospital Oakland). Bacterial cells carrying BAC-clones were electroporated with pSC101-Bad-gbaA vector. A Cre-Neo-cassette with homologous arms flanking the first exon (5′-ggctgtgaggagttacctgtcctggtacctgacaagaccacctcaccaccacttggtctc-Cre-pA-Frt-Neo-pA-Frt-tacagctgggggtcgtgtgatgggctagagggtggctaggggactgcccgatcaccaggt-3′) was electroporated into the cells to replace the first exon. Next, a targeting vector was cloned into the p15A-amp-HSV-DTA-rpsL-BSD plasmid. First two homologous arms were generated by PCR and ligated into P15A-amp-HSV-DTA-rpsL-BSD plasmid; used primers for generating homologous arms were 5′-atacggccgattggtatcaagaagcatgttagc-3′, 5′-ggtttcaaacag<u>actagt</u>tgcttcagggtataggcagcccag-3′ and 5′-taccctgaagca<u>actagt</u>ctgtttgaaacctgtgcagatatg-3′, 5′-tatgtcgacaccactggatgagggatgaggcaac-3′. Then the vector was linearized by SpeI restriction enzyme (underlined in primers) and electroporated into bacterial cells having BAC-clones with CRE-cassette in *Angpt4* locus. Colonies were picked and confirmed by sequencing. The resulting targeting vector contained 1.2 and 11.4 kb genomic arms and Cre-pA-Frt-Neo-pA-Frt cassette inserted into the first exon of the *Angpt4* gene. The AflII linearized targeting construct was electroporated into B6 ES cells and selected with G418. Genomic DNA isolated from resistant colonies was screened by PCR. The PCR assay for the targeted *Angpt4* gene used the primer pair 5′-atcactcgttgcatcgaccggtaatgc-3′ from Cre-cassette and 5′-ctgggccaggtttggttttctctg-3′ from *Angpt4* promotor region. Correctly targeted clones were injected into C57BL blastocysts, and two separate mouse lines were generated by standard methods. An Frt-flanked Neo cassette (for selection of correctly targeted ES cells) was removed from *Angpt4*<sup>frt-Neo-frt-Cre</sup> mouse line by crossing with the ACTB-FLPe transgenic mouse line. Deletion of Neo-cassette in *Angpt4*<sup>Cre</sup> was confirmed by PCR.

The ACTB-FLPe (*Rodríguez et al., 2000*) and mT/mG (*Muzumdar et al., 2007*) mice were maintained and genotyped as reported. The *Egln1*<sup>loxP</sup> mouse line was a generous gift from Professor Johanna Myllyharju (University of Oulu). The transgenic mice were maintained in the C57Bl/6N background and littermates were used as controls.

Mice were usually analyzed at four different time points that reflected early (around P4) venous development and maturation phase (around P12). Two months old mice were considered as adults (fully maturated) and aged mice (8–11 months) were used to investigate progression of retinal pathologies.

## Angiopoietin gene nomenclature

Nomenclature is based on gene naming guidelines of the HUGO Gene Nomenclature Committee (HGNC, www.genenames.org) and the Mouse Genome Informatics (MGI, www.informatics.jax.org).

## Retina preparation, whole mount staining and imaging

*Angpt4*[Cre]; *Rosa26* [mT/mG] mice eyes were fixed in 4% PFA, retinas were dissected and flattened in Immu-Mount (Thermo Scientific) between cover and objective glass for microscopy analysis. For retina immunofluorescent staining, dissected retinas were treated for 1 hr with cold methanol, permeabilized and blocked for 2 hr with 50% BSA–1% Triton–1xPBS. Primary antibody stainings were carried out overnight using Cy3-αSMA (C6198, Sigma-Aldrich), SM22 alpha (ab10135, Abcam), ColIV (AB756P, Merck Millipore), GFAP (ab7260, Abcam), pimonidazole (Pab2627, Hypoxyprobe) and biotinylated isolectin B4 (B-1205, Vector) antibodies, and visualized with Alexa Fluor 488 streptavidin, Alexa Fluor 488-, Alexa Fluor 647- and Cy3-conjugated secondary antibodies (Jackson ImmunoResearch). Retinas were imaged using Olympus FluoView FV1000 or Zeiss LSM780 confocal microscope. Coverage of αSMA-positive cells was quantified from branching point region of two largest veins at peripheral area of retina. The width of the veins was measured just below the venal branching point at peripheral retina or close to optic nerve head for comparison. 5-Bromo-4-chloro-3-indolyl β-D-galactosidase (X-Gal) staining of retinas was performed as described in *Gossler and Zachgo, 1994*.

## Quantitative RT-PCR

Total RNA was extracted by using RNeasy Mini or Fibrous Tissue Mini Kit (Qiagen) following manufacturer´s protocol. For cDNA synthesis, 1 or 3 µg of RNA was mixed with random hexamers and oligo-dT, 200 U M-MLV Reverse Transcriptase (Promega), 20 U RiboLock RNase inhibitor (Thermo Scientific), 0.5 mmol/L dNTP and reaction buffer (50 mmol/L Tris-HCl, 75 mmol/L KCl, 3 mmol/L $MgCl_2$), incubated at +42°C for 1 hr, then at +70°C for 15 min and diluted 1:3 or 1:10 in sterile $H_2O$ to be used in qPCR (2 µl/reaction). Real-time qPCR was performed using Stratagene mx3005P (Agilent Technologies) or CFX96 (Bio-Rad) qPCR instruments and Brilliant Ultra-Fast SYBR QPCR Master mix (Agilent Technologies). DNA primer sequences are indicated in the *Supplementary file 1*.

## Low oxygen treatment

Four weeks old *Angpt4*[Cre]; *Rosa26* [mT/mG] mice were exposed to low oxygen (12% $O_2$) for 8 days in a hypoxic chamber (COY Lab Products). Mice from the same litters were exposed to normoxia (21% $O_2$) as a control. Eyes were fixed in 4% PFA fixative and retinas were dissected and flat-mounted for microscopy. The relative intensity of GFP indicating the Angpt4-positive astrocytes was analyzed by ImageJ software tool.

## Oxygen-induced retinopathy (OIR) model

The OIR model was generated by exposing P7 mice to 75% oxygen in a hyperoxic chamber (controlled by ProOx Model 110, BioSpherix) for 5 days with their nursing females and then returning them to room air for 5 days. Retinas were harvested at P17.

## Hyperoxia model

Litters of P7 C57BL/6N and *Angpt4*[Cre]; *Rosa26*[mTmG] pups along with their nursing females were placed in a 75% oxygen atmosphere in a chamber (controlled by ProOx Model 110, BioSpherix) for 5 days. P12 mice were sacrificed and the eyes were enucleated. Control pups were raised under normoxic conditions and sacrificed at P12.

## Pimonidazole hypoxia staining

Vascular infusions were performed under isoflurane (1.5–2.5%) anesthesia. Before operation analgesics Karprofen (5 mg/kg) and Bubrenorfin (0.1 mg/g) were administered s.c. Pimonidazole (50 µg/µl) in 0.9% NaCl with or without 2% Evans blue was administered via femoral vein 60 mg/kg. After 1 hr, mice were sacrificed by cervical dislocation. The eyes were collected and fixed in 4% PFA for 1.5 hr at room temperature, followed by retina preparation, whole mount staining and imaging.

## Fluorosphere clearance from peripheral annular vein

Carboxylate-modified fluorescent beads (Invitrogen; 0.5 µm in diameter, excitation and emission maximum, 505/515 nm; stock concentration 2% by weigh) were diluted in 0.9% NaCl containing 3‰ Evans blue (Sigma-Aldrich) and sonicated. The left external carotid artery was permanently ligated to insert <200 µm polyethylene cannula via small arteriotomy. The cannula filled with 1% heparin

was secured with silk suture the tip residing at the site from which external carotid artery is branched. 100 µl of fluorescent beads were administered with 500 µl/min flow rate directly into common carotid artery, where it is conducted into internal carotid artery that supplies the blood to the ophthalmic artery. Mice were euthanized 27 s after the administration of fluorospheres by cervical dislocation, eyes were collected, and the distribution of fluorospheres was evaluated by confocal microcopy in whole mount preparations. For quantification, number of individual spheres was counted from images, and results were normalized by the length of venous segment analyzed.

## In vivo imaging

The animals were anesthetized by mixture of ketamine (5 mg/kg) (Ketaminol, Intervet International) and medetomidine (0.04 mg/kg) (Domitor, Orion). Both pupils were dilated with topical tropicamide (Oftan tropicamid, 5 mg/ml, Santen). To prevent corneal desiccation during procedure, topical Systane Ultra lubricant eye drop (Alcon) was applied.

## Fluorescein angiography (FA)

The mice received intraperitoneal injections of 100 µl of 5% fluorescein sodium salt (Sigma-Aldrich). Imaging was done using Heidelberg Spectralis HRA2 system (Heidelberg Engineering). Starting before the administration of sodium fluorescein a video was recorded showing the expansion of fluorescein through the retinal vascular system. In addition, snapshot images were taken every 30 s from the retinal focus level for a period of 5 min after sodium fluorescein injection. To interpret venous filling from fluorescein angiograms, fluorescein intensities were measured (ImageJ) from a video frame representing an early filling phase (arterial signal visible, but not saturated, less than 30 s from fluorescein administration). Venous intensities were normalized by an average arterial intensity from the same eye. Possible leakage was evaluated after complete filling of the retinal arteries, capillaries and veins 5 min after fluorescein administration.

## Flash electroretinography measurements (fERG)

Animals were dark-adapted for 12 hr prior to the ERG and all the ERG recordings were performed under dim red light. Mice were anesthetized as described for the in vivo imaging. The body temperature was maintained by using a physiological heating pad set to 38°C. Eyes were locally anesthetized by topical application of a single drop of oxybuprocaine (Oftan Obucain 4 mg/ml, Santen). Thereafter, eyes were fully dilated by applying drops of atropine (Oftan tropicamid 5 mg/ml, Santen) and phenylephrine hydrochloride (Oftan metaoksedrin 100 mg/ml, Santen). During the recording, eyes were kept moist with a drop of physiological saline. The a- and b-wave amplitude and latency were analysed at 0.003, 0.01, 0.1, 1, 3, and 10 cd.s/m$^2$.

## Histology and immunohistochemistry

Mice were sacrificed by $CO_2$ asphyxiation. Tissues were collected, frozen in OCT medium (Tissue Tek) and sectioned or fixed in neutral formalin and processed for paraffin sectioning. Sections of 5 µm were stained with standard Masson's trichrome, PAS or IHC methods.

## In situ hybridization

Enucleated eyes from C57BL/6N (WT) or Angpt4$^{-/-}$ mice were fixed in 4% PFA for 30 min, retinas were dissected and stored in methanol at −20°C. Upon hybridization, retinas were fixed in 4% PFA for 5 min, washed with Depc-treated PBS–0.1% Tween 20 (PBT), digested with 250 µg/ml proteinase K–0.2% SDS in PBT for 5 min and postfixed in 4% PFA–0.2% glutaraldehyde for 4 min. After brief washes with Depc-PBT and hybridization buffer (50% formamide, 5xSSC pH 4.5, 1% SDS, 50 µg/ml tRNA, 50 µg/ml heparin), retinas were hybridized with 1 µg/ml digoxigenin (DIG)-labelled mouse Tie2, Angpt1, Angpt2 or Angpt4 antisense or sense cRNA probes in hybridization buffer overnight at 65°C. Probe synthesis was performed as described previously (Powner et al., 2012). Briefly, pBluescript II SK(+) plasmids containing 1 kb Tie2 or 0.5 kb Angpt1, Angpt2 or Angpt4 cDNA (primers used for cloning are indicated in Supplementary file 2) were linearized with HindIII for antisense and BamHI for sense RNA probes, and DIG-labelled RNA probes were transcribed using T3 or T7 polymerases (Promega), respectively, and DIG-RNA labeling mix (Roche). After hybridization, retinas were first washed with 50% formamide–1xSSC for 1 hr at 65°C, then with 100 mM maleic acid

buffer containing 0.1% Tween 20 (MABT) for 40 min and blocked with 2% nucleic acid blocking reagent (Roche)–10% sheep serum in MABT for 30 min at room temperature. Retinas were incubated with anti-DIG-AP antibody (Roche) in 2% blocking reagent–1% sheep serum overnight at 4°C. After washes with MABT, retinas were briefly equilibrated in NTMT buffer (100 mM Tris pH 9.5, 100 mM NaCl, 5 mM MgCl₂, 0.1% Tween 20) and color was developed with 20 µl/ml NBT/BCIP stock solution (Roche) in NTMT–5% polyvinyl alcohol at room temperature. When staining intensity was sufficient (after 6–8 hr), retinas were thoroughly washed with PBT and fixed in 4% PFA for 1 hr at room temperature. When indicated, immunofluorescent staining was carried out as described before (*Powner et al., 2012*) by using GFAP primary antibody (ab7260, Abcam). Finally, retinas were flat-mounted and imaged with Zeiss Axio Imager motorized bright field microscope or Zeiss LSM780 confocal microscope. Overlay of confocal and bright field images was generated by ZEN (blue edition) software (Carl Zeiss). Atriums were processed otherwise similarly but they were stored in PBS and imaged with a stereomicroscope.

## Transmission electron microscopy (TEM)

Eyes were fixed in 1% glutaraldehyde and 4% formaldehyde in 0.1 mol/l phosphate buffer, pH 7.4, then postfixed in 1% osmium tetroxide, dehydrated in acetone and embedded in Epon LX112 (Ladd Research Industries). 1 µm sections were stained with toluidine blue to select regions of interest. 80 nm sections were cut with a Leica Ultracut UCT microtome and imaged using Tecnai Spirit transmission electron microscope (Fei Europe) and Quemesa CCD camera (Olympus Soft Imaging Solutions GMBH). Cellular location of β-galactosidase enzyme activity (electron dense crystals) was imaged as previously described (*Latvanlehto et al., 2010*).

## Angpt4 ELISA

Whole blood was collected from P12 mice by heart puncture immediately after sacrificing. Serum was separated by centrifugation for 10 min at 400 g at room temperature. To isolate retinal soluble extracellular and vitreous proteins, 1–2 eyes were cut open in Reagent Diluent (1% BSA in PBS, pH 7.2–7.4), sclera and lens were removed, and cellular fraction of retinas was pelleted for 5 min at 2400 g and supernatant was collected. Angpt4 level was quantified by Mouse angiopoietin-3 DuoSet ELISA kit (D4738, R & D Systems) by following the manufacturer's protocol. Absorbance at 450 nm was measured with VICTOR³V 1420 Multilabel Counter (Perkin Elmer).

## Cell culture

HUVECs (PromoCell) were cultured in M200 basal medium (Medium 200 with low-serum growth supplement, Cascade Biologics), supplemented with penicillin-streptomycin (Sigma-Aldrich) and 10% FBS (HyClone). The cell culture plates for HUVECs were coated with Attachment Factor (Cell Applications) at least for 30 min at 37°C in 5% CO₂ before seeding the cells. HUVECs transduced with full length wild type (WT) human TIE2 (TIE2-WT HUVEC), TIE2-GFP, ANGPT1, ANGPT2, or ANGPT4 were generated via retroviral gene transfer as previously described (*Saharinen et al., 2008*). Human umbilical artery (aSMC, Cell Applications) and human umbilical vein (vSMC, Cell Applications) smooth muscle cells were cultured in Smooth Muscle Cell Growth Medium (SMCGM, Cell Applications) according to the manufacturer's protocol. Human fibroblast cell line CCL-210 (ATCC) was cultured in DMEM (Gibco) supplemented with 10% FBS. All cell lines were tested mycoplasma negative when received from the manufacturer and regularly while performing the experiments.

## Astrocyte isolation and culture

Astrocytes were isolated from P3 mice brains. The brains were washed in HBSS. The cortex of the brain was dissected: Hemispheres were removed from the diencephalons and brain stem. Each hemisphere was laid on the lateral side with internal side facing up. The meninges were grasped around the olfactory bulb and pulled away gently. The dissected cortex was digested for 2 min in trypsin-EDTA (Invitrogen) and washed with HBSS. Cortexes were gently pipetted to single cell suspension and plated on poly-D-Lysin (Sigma)-coated cell culture plates. Astrocytes were grown in DMEM containing 1 g/l glucose and no L-Glutamin (Gibco), supplemented with penicillin-streptomycin (Sigma-Aldrich) and 20% FBS (HyClone). The FBS concentration was changed to 10% 48 hr after seeding. Primary cultures were stained for GFAP (ab7260, Abcam) to confirm their astrocyte identity.

### Recombinant angiopoietins

Purified recombinant human ANGPT1, ANGPT2, and ANGPT4 and mouse Angpt4 were from R & D Systems. COMP-ANGPT1 was a kind gift from Gou Young Koh.

### Fibrin gel assay

HUVECs were transduced with ANGPT1, ANGPT2 or ANGPT4 cDNAs or shRNA constructs TIE2-412, TIE2-413 or PLKO.1 control (acquired from RNAi consortium shRNA library, http://www.broadinstitute.org/rnai/trc) or left untransduced as a control. Alternatively, spheroids were stimulated with purified recombinant angiopoietins (below). Cells were suspended in SMCGM or M200 medium supplemented with 10% FBS, 1% penicillin/streptomycin, LSGS supplement and 0.32% methylcellulose, pipetted as drops to cell culture plates and incubated as hanging-drops overnight at 37°C in 5% $CO_2$ to form EC-spheroids of 750 cells and in co-culture spheroids 750 cells for each cell line. The next day, spheroids were washed with 10% FBS in 1xPBS, collected into tubes of approximately 15 spheroids and suspended in 250 μl of supplemented M200 or SMCGM medium containing 3.2 mg/ml fibrinogen and 0.32% methylcellulose and pipetted into a 48-well with 0.5 U thrombin in the bottom. After 2 hr of polymerization at 37°C in 5% $CO_2$, 300 μl of medium was added on top of the gels. Where mentioned, both the gels and the media were supplemented with 26 μg/ml serine protease inhibitor aprotinin (Sigma-Aldrich), 5 μM BYL719 (also known as alpelisib, a selective inhibitor for PI3-kinase catalytic subunit p110α, LC Laboratories) or 10 μM PI3K inhibitor LY294002 (Selleck) and/or 500 ng/ml of purified human recombinant ANGPT1, ANGPT2 or ANGPT4 or purified mouse recombinant Angpt4 (R & D Systems). After 24 hr, the cells were fixed with 4% PFA overnight at 4°C, stained with DAPI as described before (*Laib et al., 2009*) and imaged with Olympus FluoView FV1000 confocal microscope. Spheroid and lumen areas from digitized images were quantified by Olympus Fluoview Ver.3.0 Viewer. Non-stimulated HUVEC control spheroid lumen sizes were normalized to their average lumen size in independent experiments. Lumen sizes in stimulated or inhibited spheroids were normalized to non-stimulated HUVEC control lumen size within each individual experiment.

## Quantification of TIE2 activation in cell–cell junctions

TIE2-WT HUVECs were seeded on coverslips in a 24-well plate overnight. The cells were stimulated with angiopoietins (500 ng/ml) in M200 basal medium for 1 hr at 37°C in 5% $CO_2$. Cells were fixed in 4% PFA for 15 min, permeabilized in 0.5% TritonX-100 in PBS for 3 min and stained for anti-TIE2 (total TIE2, Ab33, Merck Millipore) and anti-phosphorylated TIE2 (pTIE2, Y992, Cell Signaling) antibodies. The cells were imaged with LSM780 confocal microscope (Carl Zeiss) with a Plan-Apochromat 40x/1.4 objective, and ratio of pTIE2/total TIE2 staining intensity at cell–cell junctions was quantified by using ZEN 2012 (blue edition) software (Carl Zeiss).

## Cell lysis, immunoprecipitation and western blotting

Cells were lysed in cell lysis buffer (9.1 mM Na2HPO4, 1.7 mM NaH2PO4, 1% NP-40, 0.25% sodium deoxycholate, 150 mM NaCl, 0.1% SDS, 1 mM EDTA) containing protease and phosphatase inhibitors (P8340 and P5726, Sigma-Aldrich). For immunoprecipitation, cell lysate was incubated with TIE2 antibody (sc-324, Santa Cruz Biotechnology) for 1 hr at +4°C in rotation, after which Protein A/G PLUS-Agarose (sc-2003, Santa Cruz biotechnology) was added and incubation continued overnight at +4°C in rotation, followed by washing three times with RIPA buffer. For western blot analysis, proteins were either left non-reduced or reduced by boiling for 5 min with 1% 2-mercaptoethanol (Sigma-Aldrich). Proteins were separated with SDS-PAGE and transferred into nitrocellulose membrane, followed by blocking in 5% milk powder in 0.05% Tween-PBS for 1 hr and incubation with penta-HIS (34660, Qiagen), phosphotyrosine (4G10, Merck Millipore) or TIE2 (Ab33, Merck Millipore) antibody dilutions overnight at +4°C. Primary antibodies were detected by horseradish peroxidase conjugated secondary antibodies (Jackson ImmunoResearch) with Lumi-light western blotting substrate (Roche) and signals were imaged by LAS 3000 luminescent image analyzer (Fujifilm).

## Angiopoietin binding to endothelial cell (EC) extracellular matrix (ECM)

The EC ECM fraction was generated by culturing HUVECs until confluency on coverslips, after which the cellular fraction was dissolved with 0.1% Tween-20 in PBS overnight at 4°C on a vertical shaker, and then suspended and washed vigorously until the cells were detached. His-tagged angiopoietins

(500 ng/ml) in M200 basal medium were allowed to bound to the HUVEC ECM for 1 hr at 37°C in 5% $CO_2$. Penta-His (34660, Qiagen) and HRP-conjugated secondary antibodies (Qiagen) were used to detect ECM bound angiopoietins. Washes between steps were performed with 1xPBS (with Ca and Mg), 0.05% Tween-20, pH 7.4. TMB X-tra substrate (Kem-En-Tec Diagnostics) was added for 13 min, the reaction was stopped with 0.2 M $H_2SO_4$, and the absorbance was measured at 450 nm with VICTOR[3]V 1420 Multilabel Counter (Perkin Elmer). Anti-TIE2 antibody (Merck Millipore) was used as a control for successful removal of TIE2 positive cell membrane, and anti-FN antibody (Santa Cruz Biotechnology) as a positive control for the presence of ECM in the preparations.

## Cell spreading assay on substrate-linked angiopoietins

Coverslips were first coated with penta-His antibody (2.1 µg/ml) (34660, Qiagen) overnight at 4°C, followed by blocking with 1% heat-inactivated BSA-PBS for 2 hr at room temperature. Recombinant His-tagged angiopoietins (500 ng/ml in blocking buffer) were bound overnight at 4°C. For negative controls, coverslips were only coated with penta-HIS antibody (34660, Qiagen). For positive controls, coverslips were coated with fibronectin (FN, 20 µg/ml, R & D systems). 20,000 TIE2 WT-HUVECs per coverslip were allowed to spread for 1 hr in M200 basal medium at 37°C in 5% $CO_2$. The cells were then fixed and stained with TRITC-phalloidin (P1951, Sigma-Aldrich). The cells were imaged with LSM780 confocal microscope with a Plan-Apochromat 40x/1.4 objective, and cell area was measured with ImageJ software.

## Angiopoietin–TIE2 binding affinity analysis by ELISA

96-well plates (MaxiSorp Surface, Nunc International) were coated with 4, 250 or 500 ng/ml purified human and mouse recombinant angiopoietins in PBS overnight at 4°C, followed by blocking with 1% heat-inactivated BSA-PBS (with Ca and Mg) for 1 hr. The human TIE2-Fc (250 ng/ml, R & D Systems) receptor ligand binding domain in blocking buffer was let to bind to the ligands for 15 min. Anti-human IgG Fc-tail antibody with conjugated HRP (Jackson ImmunoResearch) was used for 1 hr, TMB X-tra substrate (Kem-En-Tek Diagnostics) was added for 13 min, and the reaction was stopped with 0.2 M $H_2SO_4$. The absorbance was measured at 450 nm with VICTOR[3]V 1420 Multilabel Counter (Perkin Elmer).

## TIE2-GFP fluorescence recovery after photobleaching (FRAP)

The FRAP experiments were performed using TIE2-GFP HUVECs on glass bottomed cell culture dishes in the presence or absence of human or mouse recombinant purified angiopoietins (500 ng/ml) in environmentally controlled microscopy stage incubator. Regions of interest (ROIs) were defined and photobleached at retracting cell edges at the cell–ECM interface using the maximum laser power (405 nm) with 16 iterations. Fluorescence recovery was followed every 5 s for a period of 5 to 10 min. The following equation was used to determine the normalized fluorescence intensity (NFI):

$$NFI = \frac{\text{ROI(t)} - BG(t)}{\text{Tot(t)} - BG(t)} \; x \; \frac{\text{Tot}(t_0) - BG(t_0)}{\text{ROI}(t_0) - BG(t_0)}$$

where *BG* stands for background, *Tot* for non-bleached control ROI, and *t* for time. The plateau point was determined as a postbleach point, after which the fluorescence signal stayed constant. The mobile fraction (Mf) was determined by:

$$f = \frac{\text{NFI}_0 + \text{NFI}}{\text{NFI}_0}$$

where $\text{NFI}_0$ stands for the NFI in the prebleach period. Images were captured by a Zeiss LSM780 confocal microscope system with a Plan-Apochromat 40x/1.4 objective, and the time-lapse data was analyzed by ZEN 2012 (blue edition) software (Carl Zeiss).

## Detection of angiopoietin–TIE2 binding by surface plasmon resonance

Binding kinetics of angiopoietins to TIE2 were measured by using a Biacore T200 instrument (GE Healthcare). Purified human TIE2-Fc (R & D Systems) was immobilized to a CM5 sensor chip (5313 RU) by using standard amine coupling in 10 mM acetate buffer, pH 4. An additional channel without

coating was used in parallel as a blank channel. All samples were injected sequentially onto the sTIE2 surface at 0–200 nM concentrations in PBS using a flow rate of 30 μl/min. The 2.5 nM concentration was run in duplicate. Association time was 120 s and dissociation time 600 s. The alterations in the refractory index were recorded as relative response unit (RU).

## Cell proliferation assay

10,000 aSMC or vSMC cells per 96-well were seeded. SMCs were stimulated with angiopoietins (500 ng/ml) in SMCGM for 24–48 hr at 37°C in 5% $CO_2$. Cell proliferation was detected by Cell Proliferation ELISA BrdU kit (11647229001, Roche) following the manufacturer's protocol. Absorbance at 450 nm was measured with VICTOR³V 1420 Multilabel Counter (Perkin Elmer).

## Caspase-Glo 3/7 apoptosis assay

10,000 aSMC or vSMC cells per 96-well were seeded. SMCs were stimulated with angiopoietins (500 ng/ml) in DMEM without serum or supplements for 24–72 hr at 37°C in 5% $CO_2$. Cell apoptosis was detected by Caspase-Glo 3/7 Assay kit (G8091 Promega) following the manufacturer's protocol. Luminescence was measured with VICTOR³V 1420 Multilabel Counter (Perkin Elmer).

## Cell migration assay

aSMC or vSMC cells were grown to full confluency in SMCGM media. A wound was scratched in each well with a 120 μl tip, after which the detached floating cells were washed away with DMEM. The wound closure was followed every 15 min for 48–96 hr in the presence of angiopoietin (500 ng/ml) stimulus in DMEM with Olympus IX81 microscope equipped with motorized stage incubator and CPlanFLN PhC 10x/0.30 objective. Images were captured with Olympus XM10 CCD camera and wound closure was analyzed by ImageJ software.

## Acknowledgements

We thank Jaana Träskelin, Riitta Jokela and Anni Tenhunen for excellent technical assistance, Transgenic, Electron Microscopy and Light Microscopy Core Facilities of Biocenter Oulu for research infrastructure services, Peppi Koivunen for the access to hypoxic chamber, Nadiya Byts for astrocyte isolation from mice brains, Gou Young Koh for COMP-ANGPT1 and Johanna Myllyharju for floxed *Egln1* mouse line. This research was supported by the Academy of Finland grants to LE (251314, 136880, 310986).

## Additional information

### Funding

| Funder | Grant reference number | Author |
| --- | --- | --- |
| Academy of Finland | 251314 | Lauri Eklund |
| Academy of Finland | 136880 | Lauri Eklund |
| Academy of Finland | 310986 | Lauri Eklund |

The funders had no role in study design, data collection and interpretation, or the decision to submit the work for publication.

### Author contributions

Harri Elamaa, Conceptualization, Formal analysis, Investigation, Validation, Methodology, Visualization, Writing—original draft, Writing—review and editing; Minna Kihlström, Conceptualization, Formal analysis, Validation, Investigation, Visualization, Methodology, Writing—original draft, Writing—review and editing; Emmi Kapiainen, Conceptualization, Investigation, Visualization, Methodology, Writing—review and editing; Mika Kaakinen, Ilkka Miinalainen, Symantas Ragauskas, Marc Cerrada-Gimenez, Satu Mering, Investigation, Methodology; Marjut Nätynki, Investigation, Methodology, Writing—original draft, Writing—review and editing; Lauri Eklund, Conceptualization, Formal

analysis, Supervision, Funding acquisition, Validation, Investigation, Visualization, Methodology, Writing—original draft, Project administration, Writing—review and editing

### Author ORCIDs
Minna Kihlström (iD) http://orcid.org/0000-0002-3180-4875
Emmi Kapiainen (iD) http://orcid.org/0000-0002-7035-3544
Lauri Eklund (iD) http://orcid.org/0000-0002-3177-7504

### Ethics
Animal experimentation: Experiments involving mice were performed under permissions from the National Animal Experiment Board following the regulations of the EU Directive 2010/63/EU, the European Convention ETS123 and national legislation. License numbers ESAVI/5587/04.10.07/2013 and ESAVI/1188/04.10.07/2016.

### Decision letter and Author response
Decision letter https://doi.org/10.7554/eLife.37776.033
Author response https://doi.org/10.7554/eLife.37776.034

## Additional files

### Supplementary files
• Supplementary file 1. The sequences of primers used for quantitative RT-PCR.
DOI: https://doi.org/10.7554/eLife.37776.029

• Supplementary file 2. The sequences of primers used for generating in situ hybridization probe template plasmids.
DOI: https://doi.org/10.7554/eLife.37776.030

• Transparent reporting form
DOI: https://doi.org/10.7554/eLife.37776.031

### Data availability
All data generated or analysed during this study are included in the manuscript and supporting files.

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
