## [Decision Letter]

Thank you for submitting your article "Angiopoietin-3-dependent venous maturation and fluid drainage in the peripheral retina" for consideration by *eLife*. Your article has been reviewed by three peer reviewers, and the evaluation has been overseen by Jeremy Nathans as Reviewing Editor and Harry Dietz as the Senior Editor. The reviewers have opted to remain anonymous.

The reviewers have discussed the reviews with one another and the Reviewing Editor has drafted this decision to help you prepare a revised submission.

As you will see, all of the reviewers were impressed with the importance and novelty of your work. Reviewer #3 did have a major concern expressed in his point #3 which he guessed could take more than the usual 2 months we allow for return of a revised manuscript. Please let us know if you will be able to accommodate this recommendation.

I am including the three reviews at the end of this letter, as there are a variety of specific and useful suggestions in them.

We appreciate that the reviewers' comments cover a broad range of suggestions for improving the manuscript. Please use your best judgment in deciding which of these can be accommodated in a reasonable period of time, e.g. 2-3 months.

*Reviewer #1:*

In this manuscript, Eklund et al. describe experiments aimed to study the function of Angiopoietin-3 (Ang3) in retinal vascular development. Using mouse Knock-in technology, they produce mouse strains that allow them to detect Ang3 expression in a subset of peripheral surface astrocytes in the developing mouse retina. Targeted deletion of Ang3 coding sequences result in loss of Ang3 expression and a specific effect on retinal peripheral veins and retinal edema. The authors also describe in vitro experiments in which the activity of mouse Ang3 and its human orthologue, Ang4, is compared to that of Ang1 and Ang2. They show that although all of these related ligands function through interactions with the Tie2 receptor, they probably interact with different affinities and dynamics that result in different functionality.

This manuscript describes a large amount of elegant work. It provides the first description of Ang3 function in retinal vasculature development and the first growth factor for venous specific development. However, the manuscript contains multiple typos, redundancies, and less than clear or accurate descriptions that will have to be addressed before publication.

Below are some specific notes.

1) Is the peripheral retina in Ang3-/- mice hypoxic due to reduced circulation?

This could be addressed using Hypoxyprobe (pimonidazole HCl) injections or by monitoring the number of GFP positive cells in the retinas of Ang3cre/cre; Rosa26mTmG mutant mice. Hypoxia will result in increased number of GFP positive astrocytes.

2)The authors state that "In addition, no alterations were observed in the vasculature of the intermediate or deep vascular plexus postnatally or in adults." but provide no data to support this assertion. How was the density of the vascular plexus measured?

3) In the subsection “Ang3 deficiency results in neuronal edema”, the authors state that they did not observe cystoid edema, inflammation, or vascular leakage. The evidence for these observations is not presented or described in any details.

4) The first sentence of the Results section provides an inaccurate description of mouse retinal vascular development. Arteriovenous differentiation in the mouse retina occurs as the vessels grow on the retinal surface, beginning as early as P1 to P2.

5) Redundancies: the phrase "… human Ang4 (orthologue for mouse Ang3).…" appears at least 3 times in the figure legends.

6) Typos: Figure 1 legend, the sentence "Note that Ang3 expressed cells in Ang3+/Cre; Rosa26mTmG mice…." should read "…Ang3 expressing cells…" Also, the authors should add that all that GFP positive cells may just be progeny of Ang3 expressing cells.

7) Figure 1—figure supplement 1C, the annotation Ang3;GFP for cells in which the mTmG locus has been recombined by the Ang3cre allele seems inaccurate and confusing. It is also not used anywhere else in the text.

*Reviewer #2:*

In the manuscript entitled: Angiopoietin-3-dependent venous maturation and fluid drainage in the peripheral retina, the authors describe the generation of 3 new transgenic mouse lines targeting the Angiopoietin-4 locus and provide data on its role in activation of the tyrosine kinase receptor, Tie2 and its function in the retina. Given the importance of the angiopoietin-tie2 system, the generation of new lines targeting Angiopietin-3 is valuable. In addition, identification of venous-specific functions are intriguing and provide important new information.

A few comments for consideration by the authors are listed below:

1) The retinal phenotype is quite subtle as shown. Do the authors have any additional functional data on flow in the retina, or neural function (e.g. ERGs etc. to support a functionally significant role of Angiopoietin-4 in the retina)?

2) While the authors state that fluid drainage of the retina is affected by loss of Angiopoietin-4, it is not clear from the phenotype observed (i.e. smooth muscle effect or the neuron swelling) that this is due to poor drainage. The authors do comment on the absence of lymphatics in the retina and suggest that Angiopoietin-4 helps to promote drainage. Clarification of model and mechanisms involved leading to the phenotype would be helpful. As it is written, it is confusing to understand how the observed defects are related to 'retinal edema' and fluid drainage.

3) The smooth muscle cell phenotype is descriptive -is there anything known about potential mechanism? Is the smooth muscle phenotype only found in the periphery of veins given the expression of Angiopoietin4 in the periphery only? Also – it is not clear how the smooth muscle cell defect relates to the enlarged HUVEC diameter in vitro – is the effect on Tie2 in smooth muscle cells, on endos or both? Similar to comments in #2 above, it would be helpful if the authors provided a clear model for the reader.

4) While the Introduction talks about the importance of macula edema, it is not clear that the peripheral retinal effects would result in macula effects in humans (the location of the macula is not peripheral).

5) A variety of different timepoints are used for specific experiments – some rationale for choice of different timepoints would be useful.

6) The swollen neurons and damage to Muller cells over time is nicely shown. Were rd mutations ruled out for the mouse lines under examination? Presence of rd mutations is quite common in lab mice and can affect neuron and muller function and degeneration.

7) In the analysis of increased Angipoietin-4 expression upon PHD2 deletion – a 'Cre only' line control for comparison is missing in quantification.

8) Nomenclature of Angiopoietin3 and angiopoietin4 is confusing – official gene name in mouse is now Angiopoietin4. The authors should use the most up-to-date nomenclature.

9) The functional studies comparing various Angiopoietins is important and a strength of the manuscript – a western blot showing Tie2 phosphorylation should be performed to help support and quantify.

10) The identification of a subpopulation of astrocytes is interesting. Is there anything known about the 'subpopulation' of astrocytes expressing Angiopoietin4 – e.g. have they been identified in other studies – by RNAseq etc.? Is their origin known to be the same as other astrocytes?

11) Based on structure, is there anything known about the order of oligomerization of Angiopoietin 4 to help explain its weaker agonism of Tie2?

12) The size of the lumen appears larger after Angiopoeitin4 treatment than after Angiopietin1 treatment – is this the case? And if so, based on weaker ability to activate Tie2 – do the authors speculate an additional mechanism?

13) Is angiopoeitin4 limited to retina?

14) Is there any circulating Angiopoietin4?

*Reviewer #3:*

This manuscript describes a remarkably specific phenotype in the retinal vasculature of Ang3 KO mice. The authors show that Ang3 is expressed by retinal astrocytes during a (so far poorly characterized) vascular remodeling event that only occurs in a specific subtype of retinal veins. The findings are interesting because they demonstrate how retinal astrocytes can contribute to retinal vasculature patterning. However, the study contains several major shortcomings.

1) The mouse retinal vasculature initially develops as alternating artery-vein loops at the inner surface of the retina, i.e. there are the same number of veins as arteries. Subsequently, the deeper plexus develops and some of the veins are remodeled, ending up in the deeper plexus. Nevertheless, usually two veins stay at the surface, draining superficial flow. They bifurcate into a large annular vein at the peripheral edge of the retina (as indicated by Figure 2D). The authors should distinguish between these two types much more carefully throughout the manuscript in text as well as in figures.

2) Edema/neuronal swellinga) Only the INL was assessed in Figure 5A. However, based on the location of the vein remodeling defect, one could also anticipate reduced fluid drainage in the superficial plexus, particularly in the periphery. This should also be assessed (e.g. in the IPL).

b) It is surprising that the cellular swelling only affected the neurons and not the Müller cells (which are usually considered to be the main contributors of water drainage in the inner retina). What mechanism do the authors propose here?

c) It is also surprising that the authors did not find evidence of interstitial edema. Wouldn't one expect that from a lack of venous drainage?

d) In general, the authors should tone down their references to edema. The absence of extracellular fluid accumulation does not work well with this narrative. Furthermore, the in vivo data from the KO mice clearly points towards a vascular remodeling defect, which only secondarily leads to what may or may not be caused by reduced drainage.

3) The second part of the paper (Figure 7/8) does not link well to the first part. Firstly, the data (shown in Figure 7/8) is too superficially explained in the Results section. Secondly, I struggle to see how this relates on a mechanistic level to the phenotypes described in the KO mice and in the in vitro lumen formation assay. Yes, the authors introduce this section with the intention to demonstrate differences between Ang1 and Ang3/4 regarding Tie2 signaling, and they do show some differences. Although, the functional readout differences are not really that impressive considering Ang3/Ang4 have much lower binding affinity to Tie2 than Ang1 (Figure 7G, H). Furthermore, I fail to see how these differences relate to expansion of venous lumen in vivo, in particular when Ang1 and Ang3/4 have a similar effect (expansion of lumen) in the in vitro lumen formation assay (Figure 6).

4) Please indicate the age of the mice in Figure 2.

5) A simple, second line of evidence to demonstrate oxygen-regulated Ang3 expression (Figure 2) would be exposure to hyperoxia. Why has this not been done?

6) Please indicate the staining/visualization method in Figure 5B.

---

## [Author Response]

Reviewer #1:

[…] This manuscript describes a large amount of elegant work. It provides the first description of Ang3 function in retinal vasculature development and the first growth factor for venous specific development. However, the manuscript contains multiple typos, redundancies, and less than clear or accurate descriptions that will have to be addressed before publication.Below are some specific notes.1) Is the peripheral retina in Ang3-/- mice hypoxic due to reduced circulation?This could be addressed using Hypoxyprobe (pimonidazole HCl) injections or by monitoring the number of GFP positive cells in the retinas of Ang3cre/cre; Rosa26mTmG mutant mice. Hypoxia will result in increased number of GFP positive astrocytes.

Retinal hypoxia was investigated by using pimonidazole staining in whole mount preparations and measuring the mRNA expression levels of hypoxia-responsive genes *Angpt2* and *Vegfa* (Figure 7—figure supplement 1). We found no increase in these markers indicating that despite reduced venous function, retinas in *Angpt4^-/-^*mice are not hypoxic.

2)The authors state that "In addition, no alterations were observed in the vasculature of the intermediate or deep vascular plexus postnatally or in adults." but provide no data to support this assertion. How was the density of the vascular plexus measured?

Densities of maturated vascular plexuses were calculated from histological sections (Figure 4—figure supplement 1A-C). The pattern was evaluated from confocal microscopy data in adults (Figure 4—figure supplement 1F-G) and postnatal mice (Figure 4—figure supplement 1D-E). Ultrastructure was investigated using transmission electron microscopy in adults (Figure 4—figure supplement 1H-I). Collectively, we found no consistent changes in *Angpt4*-deficient mice in the deeper vascular plexus.

3) In the subsection “Ang3 deficiency results in neuronal edema”, the authors state that they did not observe cystoid edema, inflammation, or vascular leakage. The evidence for these observations is not presented or described in any details.

Presence of inflammation was investigated using immunohistological staining and mRNA analysis of inflammatory markers (Figure 6—figure supplement 2). In addition to marker gene analysis, general histopathology revealed no immune cell infiltration (e.g. Figure 6A-B, Figure 4—figure supplement 1A-B).

Vascular leakage was investigated by using fluorescein angiography (new Figure 7). In our study, standard clinical imaging technology had certain limitations, as the far peripheral retina cannot be imaged by using fundus camera. Therefore, vascular leakage was also investigated using periodic acid–Schiff (PAS) stained histological sections to detect hard exudates that are formed due to leakage of serum protein and lipids. Evidence for vascular leakage in PAS staining was not observed, negative result is shown in Figure 4—figure supplement 1A-B. In line with results of fluorescein angiography and PAS staining, distribution of fluorospheres (co-injected with Evans blue, a serum albumin-bound intravital dye for vascular permeability) was not suggestive for leakage (Figure 7F-G).

In cystoid edema, fluid-filled cysts disrupt the structure of retinal layers and are readily apparent in microscopy. In histological analysis (light/electron microscopy) we found no evidence for presence of alterations suggestive for cystoid edema (e.g. Figure 6A-B, Figure 4—figure supplement 1A-B and Figure 6—figure supplement 1).

4) The first sentence of the Results section provides an inaccurate description of mouse retinal vascular development. Arteriovenous differentiation in the mouse retina occurs as the vessels grow on the retinal surface, beginning as early as P1 to P2.

We have added new reference and corrected text accordingly.

5) Redundancies: the phrase "… human Ang4 (orthologue for mouse Ang3).…" appears at least 3 times in the figure legends.

Nomenclature is now corrected based on gene naming guidelines of the HUGO Gene Nomenclature Committee and the Mouse Genome Informatics. Mouse Ang3 gene is now named as *Angpt4* and protein Angpt4. Human gene is now named as *ANGPT4* and protein ANGPT4.

6) Typos: Figure 1 legend, the sentence "Note that Ang3 expressed cells in Ang3+/Cre; Rosa26mTmG mice…." should read "…Ang3 expressing cells…" Also, the authors should add that all that GFP positive cells may just be progeny of Ang3 expressing cells.

This is now corrected.

7) Figure 1—figure supplement 1C, the annotation Ang3;GFP for cells in which the mTmG locus has been recombined by the Ang3cre allele seems inaccurate and confusing. It is also not used anywhere else in the text.

This is now corrected.

Reviewer #2:

[…] A few comments for consideration by the authors are listed below:1) The retinal phenotype is quite subtle as shown. Do the authors have any additional functional data on flow in the retina, or neural function (e.g. ERGs etc. to support a functionally significant role of Angiopoietin-4 in the retina)?

Venous filling is now investigated in fluorescein angiography (FA) and using injection of fluorescent microbeads to carotid artery that supplies the blood to the ophthalmic artery. The latter approach allowed us to focus on peripheral retina that is limited in the standard FA imaging using fundus camera. In both analysis, venous flow was found to be impaired in Angpt4-deficient mice (new Figure 7).

In *Angpt4*^-/-^ mice electroretinogram (fERG) was used to measure light-induced electrical activity of the retina. We found reduction in b-wave amplitude that reflects the function of non-photoreceptors including bipolar, amacrine, and Müller cells, thus correlating with the cellular changes in the INL layer (Figure 7I).

2) While the authors state that fluid drainage of the retina is affected by loss of Angiopoietin-4, it is not clear from the phenotype observed (i.e. smooth muscle effect or the neuron swelling) that this is due to poor drainage. The authors do comment on the absence of lymphatics in the retina and suggest that Angiopoietin-4 helps to promote drainage. Clarification of model and mechanisms involved leading to the phenotype would be helpful. As it is written, it is confusing to understand how the observed defects are related to 'retinal edema' and fluid drainage.

In *Angpt4*-deficient mice, we observe defective circumferential growth of veins. Based on Poiseuille's equation, vessel flow is proportional to its radius. In line with this suggestion, functional tests using fluorescent tracers showed decreased venous function in *Angpt4*^-/-^ mice (Figure 7F-H). We propose that smooth muscle defect occurs independently from decreased venous flow as it is evident earlier than peripheral venous circulation is fully established. Alterations in neural cells are not visible in postnatal mice but appear later, suggesting that structural and functional changes in the INL occur secondary to defective venous flow and reduced venous-dependent fluid exit mechanisms. A graphical summary to clarify the model and mechanisms involved is now presented in new Figure 11.

3) The smooth muscle cell phenotype is descriptive -is there anything known about potential mechanism?

Using definite markers for SMC maturation (αSMA, SM22, myofilament structures and fibrillar collagen matrix) in different developmental stages, we demonstrate that Angpt4 is required for retinal vein SMC maturation in vivo. A limitation of our work is that we could not dissect the whole molecular pathway how Angpt4 induces SMC maturation starting from Angpt4 producing astrocytes and ending to the expression of SMCs maturation markers. In-depth in vitro mechanistical studies were hindered by the lack of SMC precursor cells and tissue culturing model that could faithfully represent SMC maturation that occurs in the retina involving endothelial cells and astrocytes.

To mechanistically better understand Angpt4 signaling pathway in vivo, we have now investigated the expression of Angpt4 receptor Tie2, as well as Angpt1 and Angpt2 in the peripheral retina at the time of venous maturation (new Figure 3A-E). Based on close location of Angpt4-expressing astrocytes to both SMCs and ECs, we consider two signaling mechanisms possible, either 1) directly from astrocytes to SMCs or 2) to SMCs via EC-produced factor. Previous studies have indicated Angpt4 receptor Tie2 expression preferentially in EC, but also in some SMC precursor cells (Iurlaro et al., J Cell Sci. 2003, 116:3635-3643). In in situ hybridization experiments, however, we found Tie2 mRNAs only in ECs (new Figure 3E). Therefore, direct signaling from astrocytes to SMCs is not likely.

Is the smooth muscle phenotype only found in the periphery of veins given the expression of Angiopoietin4 in the periphery only?

To address this question exhaustively, we investigated Angpt4 spatiotemporal expression using *in situ* hybridization in additional (earlier) developmental stages and analyzed venous αSMA expression and diameters close to optic nerve head (new panels in Figure 4, and new Figure 3). These analyses verified enhanced Angpt4 expression in the peripheral retina at P12, as we observed in genetic models. At P3 (the earliest time point we found consistent Angpt4 expression), Angpt4 was expressed in astrocytes preceding the vascular front in the mid- to peripheral retina. In contrast, Angpt4 was not expressed in central retina near to the optic nerve head (Figure 3A). In central retina, venous SMCs were not affected, thus confirming the spatial correlation of the SMC phenotype in *Angpt4*^-/-^ mice with Angpt4 expression in WT mice.

Also – it is not clear how the smooth muscle cell defect relates to the enlarged HUVEC diameter in vitro – is the effect on Tie2 in smooth muscle cells, on endos or both?

To investigate more carefully the Angpt4 signaling pathway involving ECs, SMCs and possible signaling loops between these cells, we analyzed Angpt4 receptor Tie2 expression in the retina and effects of Angpt4 in single and co-culture models of ECs and SMCs. We show that in the developing retina, Tie2 is expressed in ECs and not in SMCs (new Figure 3E). This suggested a model in which Angpt4 stimulates SMC maturation via ECs, rather than direct signaling from astrocytes to SMCs. This is in line with the results that in fibrin gel spheroid model, Angpt4-induced lumen enlargement is not enhanced by SMCs (Figure 8—figure supplement 1A), and that in single SMC culture Angpt4 did not affect SMC proliferation, survival, migration or stimulated SMC spheroids (Figure 5—figure supplement 1A-C and Figure 8—figure supplement 1B). As pointed out above, it is noteworthy that direct comparison between in vivophenotype and in vitro experiments is difficult, since currently SMC precursor cells and tissue culturing models to faithfully represent venous SMC maturation in the retina are lacking.

Similar to comments in #2 above, it would be helpful if the authors provided a clear model for the reader.

Based on new datawe proposed a model (a graphical summary in Figure 11) in which Angpt4 is produced by astrocytes, and signals via Tie2/Akt to ECs to promote venous remodeling. In *Angpt4*^-/-^ mice, venous growth and SMC maturation are defective, resulting in reduced flow, impaired fluid clearance and secondary cellular and functional defects in neuronal retina.

4) While the Introduction talks about the importance of macula edema, it is not clear that the peripheral retinal effects would result in macula effects in humans (the location of the macula is not peripheral).

Based on clinical findings, the macula can be predisposed to edema even if primary defect is peripheral retina. This is now included in the Discussion.

5) A variety of different timepoints are used for specific experiments – some rationale for choice of different timepoints would be useful.

Time points were selected based on previous knowledge regarding phases of venous development and published phenotypes in Angpt1 and Angpt2 deficient mice for comparison. Two months old mice were considered as adults (fully maturated) and aged mice were 8-11 months to investigate possible progression of retinal pathologies. This is now included in the Materials and methods section.

6) The swollen neurons and damage to Muller cells over time is nicely shown. Were rd mutations ruled out for the mouse lines under examination? Presence of rd mutations is quite common in lab mice and can affect neuron and muller function and degeneration.

*Angpt4* gene targeting was done in ES cells from C57BL/6N strain. Founder mice were crossed with ACTB-FLPe in C57BL/6J sub-strain to remove Frt-flanked neo selection gene from targeted *Angpt4* allele. Resulting *Angpt4* heterozygous mice (F1 6N /6J hybrid background) were crossed with each other to generate *Angpt4*^-/-^ and sibling control mice and were also backcrossed to C57BL/6N.

6N (but not 6J) sub-strain carries spontaneous rd8/rd8 mutation in the gene encoding *Crb1* that is normally locatedat the outer limiting membrane of the retina. Pathologies are observed *in* the outer retinal layer, including variable number of local light-colored spots due to retinal folding and slow photoreceptor loss. Pseudorosettes are apparent in histological analysis, and Müller and microglia activation and telangiectasia-like vascular lesions are reported close to the sites of retinal degeneration (Luhmann et al., Human Molecular Genetics, 2015, 24: 128-1241; Mehalow et al., Hum. Mol. Genet. 2003, 12:2179–2189; Mattapallil et al., Invest Ophthalmol Vis Sci. 2012, 53:2921-2927). Presence of rd8 mutation was detected by sequencing in our mouse colony. As expected, in mice backcrossed to C57BL/6N sub-strain, homozygous rd8/rd8 was present in all *Angpt4*^-/-^ and *Angpt4*^+/+^ mice. In 6N /6J hybrid background (F2 generation) rd8 was either present or absence in *Angpt4* deficient and control mice, that allowed us to evaluate possible effects of rd8 mutation in *Angpt4*^-/-^; Crb1^rd8/rd8^, *Angpt4*^-/-^; Crb1^+/+^, *Angpt4*^+/+^; Crb1^rd8/rd8^, and *Angpt4*^+/+^; Crb1^+/+^ genotypes. We conclude that Angpt4 and Crb1 loci are in different chromosomes (genetically not linked), rd8 and *Angpt4*^-/-^ phenotypes are clearly distinct, Angpt4 and Crb1 proteins in WT and phenotypes in deficient mice are located in the different part of retina, and that the observed phenotypes in *Angpt4*^-/-^mice are not caused or influenced by the *rd8* mutation.

7) In the analysis of increased Angipoietin-4 expression upon PHD2 deletion – a 'Cre only' line control for comparison is missing in quantification.

In Figure 2H, we compare *Angpt4* expression between *Angpt4*^+/+^; *Egln1*^loxP/loxP^ (*Phd2* is now named according its approved gene name *Egln1*) and *Angpt4*^+/Cre^; *Egln1*^loxP/loxP^ mice. In *Angpt4*^+/Cre^; *Egln1*^loxP/loxP^ mice, only one functional *Angpt4* allele remains and this has been taken into account when fold change is calculated.

In 'Cre only' i.e. *Angpt4*^Cre/Cre^, mRNAs for *Angpt4* are not transcribed (Figure 1—figure supplement 1D).

After revision was invited, we started to breed mice so that we could include P12 *Angpt4*^+/Cre^; *Egln1*^+/+^ as an additional control. Unfortunately, we failed to get litters containing necessary genotypes for comparison: *Angpt4*^+/Cre^; *Egln1*^+/+^ and *Angpt4*^+/+^; *Egln1*^loxP/loxP^ and *Angpt4*^+/Cre^; *Egln1*^loxP/loxP^.

8) Nomenclature of Angiopoietin3 and angiopoietin4 is confusing – official gene name in mouse is now Angiopoietin4. The authors should use the most up-to-date nomenclature.

In the revised manuscript, nomenclature is corrected based on gene naming guidelines of the HUGO Gene Nomenclature Committee and the Mouse Genome Informatics. This is also indicated in the Materials and methods section.

Mouse Ang3 gene is now named as *Angpt4* and protein Angpt4. Human gene is now named as *ANGPT4* and protein ANGPT4.

9) The functional studies comparing various Angiopoietins is important and a strength of the manuscript – a western blot showing Tie2 phosphorylation should be performed to help support and quantify.

We now show TIE2 tyrosine phosphorylation usingimmunoprecipitation and Western blot analysis (new Figure 9—figure supplement 2). Stimulation time was same than in immunofluorescence staining (Figure 9A) and TIE2 activation potency of mouse Angpt4 and human ANGPT4 was compared to native ANGPT1 (natural agonistic ligand), COMP-ANGPT1 (designed agonistic ligand), and Angpt2 (context-dependent antagonist/low agonist). As previously published (Lee et al., 2004), mouse Angpt4 showed a species-specific effect as it induced relatively low TIE2 tyrosine phosphorylation in human ECs when total TIE2 proteins were analyzed in WB. As shown in Figure 9A, TIE2 activation is observed in Angpt4-stimulated HUVECs as a punctuate staining specially in EC-EC junctions while a vast majority of TIE2 is evenly located on plasma membrane and not activated (negative for immunostaining for phosphorylated Tyr992 on the kinase *activation loop*). We have validated earlier the specificity of used antibody; it detects angiopoietin-stimulated TIE2-WT and over-activating TIE2-L914F mutation but does not stain TIE2 with kinase-inactivating mutation K855R (Nätynki et al., 2015). In Angpt4-stimulated HUVECs, fraction of cell-cell junction translocated and activated TIE2 is relatively small and may not be easily debatable in WB, nevertheless it is sufficient to stimulate lumen formation in a spheroid assay (Figure 8A-B). Why human ANGPT4 is more potent than mouse Angpt4 to induce TIE2 activation in human ECs is currently not known, but it may relate to lower oligomerization state of Angpt4 (Figure 9—figure supplement 2C-D).

10) The identification of a subpopulation of astrocytes is interesting. Is there anything known about the 'subpopulation' of astrocytes expressing Angiopoietin4 – e.g. have they been identified in other studies – by RNAseq etc.? Is their origin known to be the same as other astrocytes?

To best of our knowledge, Angpt4 expression has not been identified in retinal astrocytes earlier. This is likely due to lack of suitable tools to pick up relatively rare population of Angpt4 positive cells among all astrocytes. It is also currently unknown if their origin is the same as other astrocytes. The tools we have generated in this study allow isolation and characterization of Angpt4 positive astrocytes. In-depth investigation of origin, specific characteristics and importance of Angpt4 positive astrocytes requires extensive additional experiments and will be focused on in another study.

11) Based on structure, is there anything known about the order of oligomerization of Angiopoietin 4 to help explain its weaker agonism of Tie2?

We have now compared purified recombinant angiopoietin proteins used in this study in reducing and non-reducing conditions in SDS-PAGEs and Western blots (new Figure 9—figure supplement 2C-D). As expected based on previous studies, ANGPT1 formed various-sized large oligomeric/multimeric complexes, ANGPT2 ran predominantly as disulfide-linked dimers, Angpt4 showed monomeric, dimeric and some larger complexes, and ANGPT4 dimeric and higher oligomeric /multimeric ligand complexes. Interestingly, mouse Angpt4 was less potent to induce TIE2 activation in WB and this may relate to lower oligomerization state of Angpt4 (Figure 9—figure supplement 2A-D). This is now included in the Results.

12) The size of the lumen appears larger after Angiopoeitin4 treatment than after Angiopietin1 treatment – is this the case? And if so, based on weaker ability to activate Tie2 – do the authors speculate an additional mechanism?

In ANOVA (as in Figure 8A) or in t-test, Angiopoietin-4 and Angiopoietin-1 -induced lumen sizes are not different.

“do the authors speculate an additional mechanism?”: Comparison of angiopoietins has revealed that Angpt4, but not other angiopoietins, is tethered on the cell surface via heparan sulfate proteoglycans that is required for bioactivity of Angpt4 (Xu et al., J Biol Chem. 2004, 279: 41179–41188). We show here that Angpt4 and Angpt4 are not ECM bound like ANGPT1 and do not promote TIE2 translocation into EC-ECM contact sites that previously have shown to preferentially activate Erk signaling (Saharinen et al., 2008; Fukuhara et al., 2008). Differential ECM binding we found in this study may alter TIE2 signaling and bioavailability of ligands.

13) Is angiopoeitin4 limited to retina?

In this study, we primarily focused on retina as a widely used model for angiogenic growth, remodeling and maturation. As the neural retina can be considered as a part of central nervous system (CNS), we were also interested in whether Angpt4 expressing astrocytes are also present in the brains. To support this suggestion, ANGPT4 protein was reported to be expressed in normal, unselected population of human astrocytes at low level and increased in human glioma (Brunckhorst et al., 2010). However, in mouse brain astrocyte cultures we did not find Angpt4 expression (new Figure 10—figure supplement 1A). Astrocytes are known to be heterogeneous cells, and these results further strengthen our conclusion that Angpt4^+^ astrocytes represent a specific subpopulation among the glial cells.

To investigate non-central nervous tissue, we have now analyzed mesenteric whole-mounts, where vasculature is readily visible for microscopic analysis. Interestingly, Angpt4 expression was found in arterial SMCs (new Figure 10).

Negative result (no expression in brain astrocytes) and expression in certain SMCs are in line with a recent comprehensive single-cell transcriptomic study of the mouse brain vasculature (Vanlandewijck et al., 2008). Furthermore, Angpt1-GFP is reported to be expressed in the neuronal cells at P5 in mouse retina, but also in perivascular mural cells in choroidal vessels. The data is currently somewhat limited but suggests that in non-retinal tissue expression of Angpt1 and Angpt4 may be overlapping in some perivascular mural cells. This raises interesting questions regarding specific/synergistic/additional effects of these Tie2-agonistic ligands in non-CNS tissues that will be addressed in subsequent studies.

14) Is there any circulating Angiopoietin4?

The presence of Angpt4 was invested in P12 serum using commercially available ELISA. Based on this assay, there is no detectable Angpt4 in mouse sera (new Figure 10—figure supplement 1B). In addition, amount of soluble Angpt4 in eye preparation was relatively low.

Reviewer #3:

This manuscript describes a remarkably specific phenotype in the retinal vasculature of Ang3 KO mice. The authors show that Ang3 is expressed by retinal astrocytes during a (so far poorly characterized) vascular remodeling event that only occurs in a specific subtype of retinal veins. The findings are interesting because they demonstrate how retinal astrocytes can contribute to retinal vasculature patterning. However, the study contains several major shortcomings.1) The mouse retinal vasculature initially develops as alternating artery-vein loops at the inner surface of the retina, i.e. there are the same number of veins as arteries. Subsequently, the deeper plexus develops and some of the veins are remodeled, ending up in the deeper plexus. Nevertheless, usually two veins stay at the surface, draining superficial flow. They bifurcate into a large annular vein at the peripheral edge of the retina (as indicated by Figure 2D). The authors should distinguish between these two types much more carefully throughout the manuscript in text as well as in figures.

In whole mounts, these two vein types were distinguished based on presence or absence of connection to the large annular vein at the peripheral retina. Most analysis were done close to bifurcation site of the large annular vein and ora serrata. In histological sections, we focused on far peripheral region of retina so that peripheral and paracentral veins were not mixed. Exceptions are Figure 7 (due to limited field of view in fluorescein angiography only shortest veins were excluded, analysis may contain few veins that are extended to more periphery but not connected to annular vein in the far periphery) and Figure 4H and K (all centrally located veins were included for comparison vs. peripheral veins). Different types of veins (peripheral annular and central/paracentral) are now distinguished in the figures whenever possible.

2) Edema/neuronal swellinga) Only the INL was assessed in Figure 5A. However, based on the location of the vein remodeling defect, one could also anticipate reduced fluid drainage in the superficial plexus, particularly in the periphery. This should also be assessed (e.g. in the IPL).

We have now carefully analyzed additional retinal layers (Figure 6—figure supplement 1). Ultrastructural changes suggestive for neuronal cell swelling (increased volume of cell organelle free cytoplasm, disordered cellular organization) were observed in the INL, but not in other retinal layers. Defective vein development in *Angpt4*^-/-^ mice shares some similarities with retinal vein occlusions (RVO) (i.e. reduced venous drainage) but is less severe and does not result in inflammation or leakage that further increases excessive fluid in RVO. In the peripheral retina Angpt4 receptor Tie2 is expressed only in endothelial cells (new Figure 3E), suggesting that direct Angpt4 signaling from astrocytes to neuronal cells is unlikely. Furthermore, we observe alterations in the INL in mature eye after venous remodeling defect and high Angpt4 expression at P12. Based on our data, we propose that ultrastructural changes in the INL are secondary to reduced vein function. Distance between the superficial vasculature and the INL is about 50 μm (roughly equals with the average distance between capillaries in the retina). Functional venous flow may be especially important for INL, while diffusion to the vitreous in the most superficial layer and pigment epithelial cell pumps in outer layer may provide additional/compensatory mechanisms. Schematic model is presented in new Figure 11.

b) It is surprising that the cellular swelling only affected the neurons and not the Müller cells (which are usually considered to be the main contributors of water drainage in the inner retina). What mechanism do the authors propose here?

In electron microscopic analysis of INL, we observe changes in the normal architecture of Müller cells (discontinuation of Müller cell processes)that normallyspan across the retina, but no increase in Müller cell volume. Swelling of Müller cells occur in retinal pathologies and associated altered expression of AQP4 water or Kir4.1 potassium channel in Müller cells that may result in dysfunctional water homeostasis in these cells. In *Angpt4*^-/-^ retinas, we found no changes in AQP4 nor Kir4.1, suggesting that water balance in Müller cells may be retained. Neuronal cells (amacrine and bipolar cells in the INL), may lack efficient water transport mechanism and be more vulnerable for alterations in fluid homeostasis due to diminished venous drainage. This can be aggravated by the alterations in Müller cell architecture. This assumption is now included in Discussion.

c) It is also surprising that the authors did not find evidence of interstitial edema. Wouldn't one expect that from a lack of venous drainage?

In the revised manuscript, we investigated the severity of venous dysfunction in *Angpt4*^-/-^ retinas. In fundus angiography, venous filling is slower in *Angpt4*^-/-^ mice and carotid artery injected fluorospheres accumulated in the peripheral annular vein in the *Angpt4*^-/-^ mice (new Figure 7). Based on these functional assays, venous drainage is impaired, but not completely blocked. As also pointed above, in literature we found no similar pathology than we observe in *Angpt4*^-/-^ mice (i.e. decrease in vein diameter, diminished peripheral vein flow, venous SMC abnormality, no inflammation, no leakage), making direct comparison to other retinal vascular pathologies (i.e. diabetic retinopathy, retinal vein occlusion) difficult. It may be possible that interstitial edema is prominent in vascular diseases in which vascular barrier disrupted, which causes large increase in fluid entry that is excessive for exit mechanisms. Those neuronal cells that lack efficient water pumping mechanism may make them more vulnerable for passive influx of fluid that may be increased in *Angpt4*^-/-^ retinas. This is now discussed.

d) In general, the authors should tone down their references to edema. The absence of extracellular fluid accumulation does not work well with this narrative. Furthermore, the in vivo data from the KO mice clearly points towards a vascular remodeling defect, which only secondarily leads to what may or may not be caused by reduced drainage.

This is done. In *Angpt4*^-/-^ mice, we first observe venous remodeling defect in postnatal mice and we propose that other phenotypes (impaired venous drainage and neuronal cell changes) are secondary. The model is now presented in schematic summary (new Figure 11).

3) The second part of the paper (Figure 7/8) does not link well to the first part. Firstly, the data (shown in Figure 7/8) is too superficially explained in the Results section. Secondly, I struggle to see how this relates on a mechanistic level to the phenotypes described in the KO mice and in the in vitro lumen formation assay. Yes, the authors introduce this section with the intention to demonstrate differences between Ang1 and Ang3/4 regarding Tie2 signaling, and they do show some differences. Although, the functional readout differences are not really that impressive considering Ang3/Ang4 have much lower binding affinity to Tie2 than Ang1 (Figure 7G, H). Furthermore, I fail to see how these differences relate to expansion of venous lumen in vivo, in particular when Ang1 and Ang3/4 have a similar effect (expansion of lumen) in the in vitro lumen formation assay (Figure 6).

In *Angpt4*^-/-^ mice, we observed venous SMCs abnormality and smaller diameter of annular veins. Possible molecular and cellular pathway by which Angpt4 may promote luminal growth and EC to SMC signaling were investigated in vitro either as single cell type or in co-culture models. Another motivation for detailed in vitro comparison of angiopoietins was that it has been controversial how well human and mouse Angpt4functions are conserved due to divergence in their primary structures. Reviewer 2 foundstudies comparing angiopoietins important and strengthening the manuscript. In addition, we were also requested for some additional in vitro comparisons of angiopoietins that we have faithfully done (Figure 9—figure supplement 2). Please find below, how in vitro and in vivo experiments are linked together, and how in vitro experiments help mechanistically understand in vivo results. Also, the data is now explained more extensively in the Results section.

Tie2 is mainly expressed in ECs, however, its expression is also reported in vascular mural cell precursors (Iurlaro et al., J Cell Sci. 2003, 116:3635-3643), and therefore we considered it possible that astrocytes may directly signal to SMCs. In line with this suggestion we found some TIE2 expression in SMC lines isolated from maturated veins and arteries (Figure 5K). Excesses of ANGPT4, however, had no effect on SMCs cellular functions (proliferation, migration, survival) (Figure 5—figure supplement 1).

As SMC lines isolated form maturated veins may not well represent SMC precursor cells in the developing retina, we further investigated Angpt4 receptor *Tie2* mRNA expression in retinas using in situ hybridization at P12, when Angpt4 expression is high and venous remodeling of large annular vein occurs. As shown in new Figure 3E, *Tie2* expression is detected only in ECs in peripheral retina. Collectively, based on in vitro and in vivo data, direct Angpt4 signaling from astrocytes to SMCs via Tie2 is not likely.

Possible molecular pathway by which ANGPT4 may increase expansion of luminal structure was investigated in fibrin gel model, either using ECs alone or co-culture model of EC/SMC spheroids (Figure 8 and new Figure 8—figure supplement 1A). ANGPT4 and Angpt4 expanded lumen structure in ECs and addition of SMCs had no effect, also proposing that ANGPT4 and Angpt4 may primarily signal to ECs via TIE2. In line with this suggestion, TIE2 shRNA silenced HUVECs did not respond to ANGPTt4 in a spheroid assay, indicating the dependence of endothelial TIE2 in expansion of luminal structure in vitro.

Comparison of ANGPT1, Angpt4 and ANGPT4 in the spheroid assay revealed that despite their divergent primary structures and differences in biochemical properties, we found that they all are competent to increase luminal structure via the same signaling pathway. In addition, careful analysis of EC-EC junctions revealed that they all activated TIE2 in this specific subcellular compartment that is important for TIE2/AKT signaling (Saharinen et al., 2008; Fukuhara et al., 2008). These results suggest that functions of ANGPT1, Angpt4 and ANGPT4 are partly conserved. This is in line with a previous study in mouse, in which excess of these ligands similarly induced blood and lymphatic vascular remodeling (Kim et al., 2007).

In the far peripheral retina, Angpt2 and Angpt4 (new Figure 3) were both expressed at P12 when venous maturation occurred in this domain. in vitro, ANGPT2 did not induce TIE2 activation in EC-EC junctions and did not enhance lumen formation, indicating that ANGPT2 less potent to functionally compensate ANGPT4 to promote venous growth, thus partly explaining specific phenotype in Angpt4 deficient mice.

Collectively, combination of in vitro and in vivo results suggested a model in which Angpt4 primarily signals via endothelial Tie2 to enhance EC to SMC signaling for SMCs maturation (new Figure 11). The previously observed vascular phenotype in Angpt4 deficient retinas was clearly different from that observed in mice lacking Angpt1 or Angpt2 and based on our data we propose that differentially regulated expression of angiopoietins is an important mechanism to generate ligand-specific functions in the developing eye.

ANGPT1 is an obligatory agonistic ligand and binds to TIE2 with a high affinity (Kd 3.7 nM, Davis et al., Cell, 1996, 8:1161–1169). We show that although binding affinity of human ANGPT4 to TIE2 is lower, it is still sufficient for TIE2 activation in Western blot and in cell-cell junctions. In this specific compartment, also mouse Angpt4 activated human TIE2.

4) Please indicate the age of the mice in Figure 2.

This is done.

5) A simple, second line of evidence to demonstrate oxygen-regulated Ang3 expression (Figure 2) would be exposure to hyperoxia. Why has this not been done?

We were originally interested in ischemia that is considered to be more relevant for many vascular diseases. *Angpt4* gene promoter contains hypoxia response element and based on our data we are confident that *Angpt4* expression in positively regulated by low oxygen. Nevertheless, hyperoxia experiment is now done; WT and *Angpt4*^Cre^ ; *Rosa*26^mTmG^ mice were placed in a 75% oxygen chamber for 5 days and euthanized at P12 together with control mice that were raised under normoxic conditions. As shown in new Figure 2—figure supplement 2, *Angpt1* mRNA expression was decreased while *Angpt2* and *Angpt4* increased. In addition, there was a trend for increased number of *Angpt4*^+^ astrocytes. The results are not straightforward to interpret as raising newborn mice in hyperoxia may not be a simple negative control for hypoxia: Hyperoxia has reported to paradoxically cause retinal hypoxia by blocking vascular development (West et al., Development, 2005) that may perhaps explain increase in *Angpt2* and *Angpt4* expression. In addition, hyperoxia has toxic effects in the retina causing retinal degeneration (Yamada et al., J Cell Physiol. 1999) and it may be possible that increase in *Angpt4* may relate to more general stress reaction in the retina. Finally, a short-term hyperoxia promotes astrocytic differentiation (Duan et al., Sci Reports, 2017) and may also affect *Angpt4*^+^ astrocytes. In addition to hypoxia, other mechanism to regulate *Angpt4* expression also likely exists.

6) Please indicate the staining/visualization method in Figure 5B.

This is now done.